

# Precipitation and ice core δD-δ¹⁸O line slopes and their climatological significance

Ben G. Kopec[1,2], Xiahong Feng[2], Erich C. Osterberg[2], Eric S. Posmentier[2]

[1]Department of Geology, Gustavus Adolphus College, St. Peter, MN, 56082, USA
5   [2]Department of Earth Sciences, Dartmouth College, Hanover, NH, 03755, USA

*Correspondence to*: Ben G. Kopec (bgkopec@gmail.com)

30





**Abstract.** The meteoric water line, defined by the correlation of hydrogen ($\delta D$) and oxygen ($\delta^{18}O$) values, is one of the earliest described characteristics of precipitation isotopic variations. However, spatial and temporal variations in the slope of this line are less studied. The slope of the $\delta D$-$\delta^{18}O$ relationship is coupled with how d-excess covaries with $\delta D$ or $\delta^{18}O$, and may provide an integrated tool for inferring hydrologic processes from the evaporation to condensation site. We present a study of $\delta D$-$\delta^{18}O$ relationships on seasonal and annual timescales for event-based precipitation and a 15-meter ice core (Owen) at Summit, Greenland. Seasonally, precipitation $\delta D$-$\delta^{18}O$ slopes are less than eight (summer=7.71; winter=7.77), while the annual slope is greater than eight (8.27). We suggest intra-season slopes result primarily from Rayleigh distillation, which, under prevailing conditions, produces slopes less than eight. The summer line has a greater intercept (higher d-excess) than the winter line. This separation causes annual slopes to be greater than seasonal ones. We attribute high summer d-excess to contributions of vapor sublimated from the Greenland Ice Sheet. Higher sublimated moisture proportions in summer cause larger separations between seasonal $\delta D$-$\delta^{18}O$ lines, and thus higher annual slopes. Intra-seasonal distributions of precipitation amount also influence annual slopes because slopes are weighed by the number of storms each season. We generate indices to quantify sublimation proportion (SPI) and precipitation distribution (PDI), and find that annual Owen core slope measurements are significantly related to these indices, demonstrating that sublimation and precipitation distribution represent important climate conditions recorded in ice cores.

## 1 Introduction

Over the past half century, variations of hydrogen ($\delta D$) and oxygen ($\delta^{18}O$) isotopic ratios of precipitation have served as increasingly powerful tools in a wide range of disciplines of research, including paleoclimate, hydrology, and atmospheric sciences. One of the most striking features of these two paired isotope ratios is that they are remarkably well correlated over time and space, and this relationship is defined as the meteoric water line (MWL). This line is among the earliest described characteristics of the isotopic distribution in global precipitation, and worldwide, the MWL has the form $\delta D = 8*\delta^{18}O + 10$ (Craig, 1961). For a set of precipitation samples at a given location, the line defining the $\delta D$-$\delta^{18}O$ relationship may deviate slightly from this global mean, and is often referred to as the local meteoric water line, or LMWL. While most of the variance in $\delta D$ and $\delta^{18}O$ values is along the LMWL, there are samples whose values deviate from this line. These deviations are often expressed by the deuterium excess (d-excess), defined as the difference $\delta D - 8*\delta^{18}O$ (Dansgaard, 1964). Measurements of $\delta D$, $\delta^{18}O$, and d-excess have yielded extraordinary information about the climate system. In the field of paleoclimate studies, for example, $\delta D$ or $\delta^{18}O$ variations in ice cores have been used to infer temperature changes at ice core drilling sites (e.g. Jouzel et al., 1987; Dansgaard et al., 1989; Johnsen et al., 1995, 2001; Blunier et al., 2001; Petit et al., 1999) and deuterium-excess variations in ice cores have been used to infer changing marine moisture source conditions over a range of time scales (e.g. Johnsen et al., 1989; Barlow, et al., 1993; Vimeux et al., 1999, 2001; Uemura et al., 2004, 2012; Masson-Delmotte, et al., 2005; Jouzel et al., 2007).



There are relatively fewer studies focusing on how the slope of the MWL changes over space and time, and determining how such variations may contain climate information. Since d-excess is computed from $\delta D$ and $\delta^{18}O$, there are close relationships among their variations. For example, if the slope of a LMWL with seasonally resolved observations is less than 8, then d-excess and $\delta D$ (or $\delta^{18}O$) would have an anti-phase relationship, and the opposite is also true. Feng et al.

(2009) reported that in the majority of northern temperate and Arctic locations, $\delta D$ reaches a seasonal maximum when d-excess reaches its minimum. This out-of-phase relationship between the two is equivalent to the fact that, in these locations, the LMWL (with a monthly or higher resolution) has a slope less than 8. The authors attributed this seasonal d-excess cycle to the north-south migration of subtropical moisture source areas over the year, so that vapor from marine moisture sources in the summer is evaporated from more northern locations with cooler sea surfaces and higher relative humidity as compared

to those in winter. To our knowledge, this is the only work that described the slope distribution of LMWLs on a hemispheric scale and discussed the climatological significance of these slopes. Temporal changes, e.g., seasonal or inter-annual, in the $\delta D$-$\delta^{18}O$ relationship for a given location have not been explored and can potentially provide information and understanding of seasonal or interannual variations in the planet's climate system.

Summit, Greenland is one of the most important sources of deep ice cores that provide valuable paleoclimate

records, particularly through the measurement of water isotopes. Despite the depth of information we have learned from these measurements in ice cores, a recent study has shown that there may be more to the story than we presently understand, particularly regarding the deuterium excess records. While Feng et al. (2009) demonstrated that almost all sites in the mid- to high-latitudes of the Northern (and Southern) hemisphere exhibit an out-of-phase relationship between $\delta D$ and d-excess, and the mechanisms controlling this pattern are relatively well understood, Kopec et al. (2019) recently reported a nearly in-

phase relationship between $\delta D$ and d-excess of event-based precipitation measurements at Summit, Greenland. This unique relationship requires additional investigation into the mechanisms controlling the water isotope record, primarily d-excess, at Summit. In addition to changing marine moisture sources as a major control of d-excess, to explain the unique d-excess variations at Summit, Kopec et al. (2019) discussed two mechanisms: 1) Rayleigh distillation and 2) sublimation as a source of water vapor for precipitation. Rayleigh distillation alters d-excess through a nonlinear evolution of the $\delta D$-$\delta^{18}O$ trajectory

as moisture in an air mass progressively condenses and rains out (Craig, 1961; Majoube, 1971; Ciais and Jouzel, 1994; Dutsch et al., 2017). Kopec et al. (2019) demonstrated that Rayleigh distillation can help explain the relatively low d-excess values in winter Summit precipitation. On the other hand, Summit d-excess in the summer is extraordinarily high, which they explained via moisture contribution from sublimation of surface snow on the Greenland Ice Sheet along the air mass transport path. Water vapor sourced from sublimation has a relatively high d-excess and Kopec et al. (2019) demonstrated

that this mechanism provides a significant proportion of moisture to summer precipitation at Summit. They considered sublimation to be especially important for ice core interpretations because this vapor provides a high d-excess moisture source, in addition to vapor evaporated from a warm sea surface, to the total precipitation.

The unique phase of the annual cycle of d-excess brings up an additional factor for interpreting ice core records - the seasonal distribution of precipitation. Any change in the seasonality of precipitation at Summit, such as when winter





precipitation at Summit was hypothesized to be reduced significantly during the Last Glacial Maximum (Werner et al.,
2000), was previously interpreted to have minimal effect on ice core d-excess records. This was because winter d-excess
values were thought to be similar to that of the annual average, with seasonal extremes in the spring and fall months
(Masson-Delmotte et al., 2005). However, this newly identified annual cycle, where the seasonal extremes are in winter and
summer, suggests that a change in precipitation seasonality will greatly impact the d-excess observed in these records.

In this study, we investigate temporal changes in the slope of the δD-δ¹⁸O relationship at Summit, Greenland. We
observe these variations from event-based precipitation and a shallow ice core (Owen) collected at Summit. We focus on
understanding the role of the sublimation moisture source and the seasonality of precipitation in defining the slope of the δD-
δ¹⁸O line, and how temporal changes in these factors are recorded in ice cores. The specific objectives of this study are 1) to
determine the slopes of δD-δ¹⁸O lines over seasonal and/or interannual timescales in precipitation and an ice core, 2) to
examine the hypothesized effect of sublimation-sourced moisture and the seasonality of precipitation on these slopes, and 3)
to explore implications for isotopic interpretation of deep ice cores. For convenience, we will refer to the slope of the δD-
δ¹⁸O line as the δD-δ¹⁸O slope or simply the slope.

## 2 Methods

### 2.1 Precipitation collection

Precipitation samples were collected daily at Summit, Greenland (72.58°N, 38.48°W) from July 2011 through
September 2014. Snow was captured by free hanging nylon bags at heights of 1, 2, and 4 m from the ground surface. For this
work, we used only samples taken at the top height (4 m), which have been shown to be largely free of contamination by
blowing snow and thus represent only new precipitation (Kopec et al., 2019). Details of the sampling procedure are
described in Kopec et al. (2019).

### 2.2 Owen ice core collection

The Owen ice core was collected at Summit, Greenland in June 2010. The ice core was drilled to a depth of 100 m
using the four-inch drill from the U.S. Ice Drilling Program Office (Hawley et al., 2014). The core was shipped frozen to the
Dartmouth Ice Core Laboratory where the top 15 meters was logged, subsampled, and analyzed for water isotopic
composition.

### 2.3 Isotopic analysis and ice core dating

The isotopic analysis of the precipitation was conducted at the Stable Isotope Laboratory at Dartmouth College.
Hydrogen isotopic ratios (δD) were measured using an H-Device, in which water is reduced by hot chromium (850°C) and
the resulting $H_2$ gas is measured by an isotope ratio mass spectrometer (IRMS, Thermo Delta Plus XL). Oxygen isotopic
ratios (δ¹⁸O) were determined using the $CO_2$ equilibration method with a Gas Bench interfaced with the IRMS. Deuterium





excess (d or d-excess) was calculated as d = δD – 8*δ$^{18}$O. The analytical uncertainty (1σ) is 0.1‰ for δ$^{18}$O, 0.5‰ for δD, and 0.9‰ for d-excess.

The Owen ice core was subsampled for stable isotope analysis using the continuous ice core melting system with discrete sampling at the Dartmouth Ice Core Laboratory (Osterberg et al., 2006). In total, 249 samples were collected at 4 -

11 cm intervals. The hydrogen (δD) and oxygen (δ$^{18}$O) isotopic ratios of melt ice were measured using a Picarro L2130-I cavity ringdown analyzer. The analytical uncertainty (1σ) is 0.2‰ for δ$^{18}$O, 0.6‰ for δD, and 1.7‰ for d-excess.

The Owen ice core was dated from the end of 1977 to the beginning of 2010 primarily using δ$^{18}$O variations. The annual cycle of the isotopic ratios is quite clear with winter minimum and summer maximum values. The dating was confirmed further using annual cycles of major ions including SO$_4$$^{2-}$, Ca$^{2+}$, and Na$^+$.  January 1$^{st}$ of a given year was defined as the δ$^{18}$O

minimum. We use this method for each year except for the start of 1982 and 1987, when winter minimums are poorly defined or missing. We identified January 1$^{st}$ of these two years based on the major ion annual cycles. The number of measurements in each year ranges from 4 to 11. The slope of the δD-δ$^{18}$O line was determined for each year using a simple linear regression of all data points within a single year from the minimum value (January) to the last data point before the next minimum (~December). Using this same annual delineation, we also calculate the thickness of each annual layer.

**2.4 Correction for isotope diffusion in the ice core**

Diffusion and vapor ice exchange as the snow or firn compacts and ages may alter the slope of the δD vs. δ$^{18}$O relationship. With diffusion, the contrast of isotopic ratios of the seasonal extremes (e.g., summer and winter) reduces over time, and eventually the annual cycle is removed completely (Cuffey and Steig, 1998; Johnsen et al., 2000). Cuffey and Steig (1998) modelled the amplitude of the isotopic annual cycle for 350 years of the GISP2 ice core from Summit, and showed

that the amplitude could be reduced to approximately half of the initial value after ~30 years. Because oxygen isotopic ratios change faster than hydrogen isotopic ratios (Merlivat, 1978; Johnsen et al. 2000), the reductions of the amplitudes are not equal, which alters the d-excess values and the slope of the δD-δ$^{18}$O line. Johnsen et al. (2000) showed that this effect caused an increase of the δD vs. δ$^{18}$O slope from eight at the surface to 11 in deeper firn of about 100 years of age.

We used the diffusion model from Johnsen et al. (2000) to correct the effect of diffusion on the calculated δD-δ$^{18}$O

slope. The model has the form

$$A^i = A_0^i \exp(-\tfrac{1}{2}k^2\sigma_i^2) \tag{1}$$

where $A^i$ is the isotopic amplitude of the annual cycle of either δ$^{18}$O ($i$ = 18) or δD ($i$ = D), $A_0^i$ is the initial amplitude of the

annual cycle, $k$ is the wave number, and $\sigma_i$ is the diffusion length of the given isotope ratio. The wave number is defined as $2\pi/\lambda$, where $\lambda$ is the thickness (cm) of an annual layer. The diffusion length, $\sigma_i$ (cm), depends on a number of factors including vapor diffusivity in the firn pore space, equilibrium isotopic fractionation factor between ice and vapor, time, and



the strain rate of compression that is, among other factors, related to the snow accumulation rate (Johnsen et al., 2000). It is the relationship of a given water isotopologue with the vapor diffusion rate and equilibrium isotopic fractionation that results in different values of diffusion length for each water isotopologues. The diffusion length for each isotope was calculated by Johnsen et al. (2000) for the top 150 m of firn at Summit, Greenland, and is used in this work.

5 The slope of a measured annual cycle is equivalent to $A^D / A^{18}$, while the original slope of the annual cycle at the time of deposition is $A_0^D / A_0^{18}$. In order to determine the original slope prior to diffusion, we rearrange versions of Eq. 1 for both isotope ratios and combine the two to obtain

$$\frac{A_0^D}{A_0^{18}} = \frac{A^D}{A^{18}} \exp\left[\frac{1}{2}k^2(\sigma_D^2 - \sigma_{18}^2)\right] \tag{2}$$

where the change of slope is forced by the difference of the squared diffusion lengths. We use this equation to calculate the original slope ($A_0^D / A_0^{18}$) of the δD-δ¹⁸O line each year. In order to solve this equation, we measure the slope as $A^D / A^{18}$, we calculate the wave number $k$ using the measured annual layer thickness, and we obtain the respective isotope ratio diffusion lengths from the calculations by Johnsen et al. (2000).

15 **3 Results**

We focus on the variations in the slope of the annual δD-δ¹⁸O line at Summit, Greenland and their climatological significance. We first report the slopes of event-based precipitation isotopic ratios on seasonal and annual time scales. We then describe the shallow ice core slope variations, including both the measured data and the data corrected for diffusion.

**3.1 Precipitation**

20 For the entire data set as a whole, as well as within each calendar year, the slope of the δD-δ¹⁸O line is greater than 8 (Fig. 1). The full data set has a slope of $8.27 \pm 0.05$ (mean ± standard error), and 2012 and 2013 have slopes of $8.36 \pm 0.08$ and $8.06 \pm 0.09$, respectively. This result is consistent with the observations of Kopec et al. (2019), where they reported d-excess measurements in phase with δD or δ¹⁸O values.

On a seasonal timescale, δD-δ¹⁸O lines exhibit different patterns, and herein we focus primarily on the differences 25 between summer and winter δD-δ¹⁸O slopes. The average slopes of the summer (June, July and August; JJA) and winter (December, January, and February; DJF) lines are $7.71 \pm 0.06$ and $7.77 \pm 0.10$, respectively (Fig. 1a). For reference, the spring (March, April, and May) and fall (September, October, and November) δD-δ¹⁸O lines have slopes of $8.28 \pm 0.09$ and $7.71 \pm 0.09$, respectively. The slopes of the summer and winter δD-δ¹⁸O lines are not significantly different (p = 0.60), but the two lines have significantly different intercepts. The summer intercept ($7.30 \pm 1.72$) is significantly greater (p = 0.0009)



than the winter one (–6.28 ± 3.89), which shows that the summer line is essentially parallel with the winter line but is elevated above that line.

The seasonal difference between winter and summer is more easily discerned when observing the data in d-excess-$\delta^{18}$O space (Fig. 1b). A $\delta$D-$\delta^{18}$O line with slope of eight would plot in this space as a horizontal line; $\delta$D-$\delta^{18}$O lines with

slopes greater (less) than eight would plot in this space with positive (negative) slopes. Winter and summer lines both have similar negative d-excess-$\delta^{18}$O slopes, and the summer line is clearly above the winter line with higher d-excess values. A second observation clearly seen in Fig. 1 is that the annual $\delta$D-$\delta^{18}$O slope is greater than eight while winter and summer slopes are less than eight. The seasonal separation of the summer line above the winter line (or higher summer d-excess than winter d-excess) results in a slope greater than eight when data from all seasons are included in the linear regression.

Without this separation, the overall regression should have yielded a slope similar to that of summer or the winter lines.

**3.2 Owen ice core**

The values of $\delta^{18}$O and d-excess for the Owen ice core are shown in Fig. 2. In general, d-excess variations are close to being in phase with that of $\delta^{18}$O. The only exception to this pattern occurs near the top of the core, from 2008 to 2010, where d-excess appears to be anti-phase with oxygen isotopic ratios.

Without correction for diffusion, the slope of $\delta$D vs. $\delta^{18}$O of all data from 1978-2009 is 8.49 ± 0.05, which is consistent with the fact that the isotopic ratio and d-excess are largely in phase (or at least more in-phase than out-of-phase). The annual slopes are plotted as a function of time in Fig. 3a. The average of the individual slopes is 8.67 and ranges from 7.48 to 9.68, with 27 out of 32 slopes greater than eight. Over time, the slopes decrease significantly (Fig. 3a; $r^2$ = 0.18; p = 0.016), and the annual slopes of the precipitation data of 2011 to 2014 appear to fit this trend. There are three missing data

points due to unsuccessful isotopic measurements, one each from 1982, 1983, and 1997. This could have affected the computed slope for the regression of these years, especially in the case of 1997, where the missing isotopic value would likely be the maximum of the year. However, removing these years does not change any of the above analysis, within error.

Correction for the diffusion effect on the isotopic record produces some significant differences, most prominently, the elimination of the temporal trend in the $\delta$D-$\delta^{18}$O slope. Using the model described in 2.4, the slope at the bottom layer of

the ice core (year = 1978; 33 years before drilling) was reduced from the measured value of 8.91 to the corrected value of 7.73. With all slopes adjusted for diffusion, the corrected annual slopes of $\delta$D-$\delta^{18}$O lines are replotted in Fig. 3b. Since slopes of earlier years are subject to greater corrections, the decreasing temporal trend of the slopes is no longer statically significant (p = 0.23). After the correction, 18 of 32 $\delta$D-$\delta^{18}$O annual slopes are greater than eight.

**4 Discussion**

In the prior sections, we have determined variations of slope of the $\delta$D-$\delta^{18}$O line for event-based precipitation collected at Summit, Greenland over seasonal and annual timescales. We also determined interannual slope variations from a



high-resolution ice core. In what follows, we identify and discuss factors that control the seasonal and annual slopes of δD-δ¹⁸O lines, quantify the effect of these factors, and discuss their implications for ice core interpretation.

### 4.1 Controls of precipitation δD-δ¹⁸O slopes

Craig (1961) first observed that the correlation line for δD and δ¹⁸O values of meteoric water samples across the

planet have a slope of 8, and an intercept of 10‰, which was defined as the meteoric water line (MWL). The slope of the MWL was considered largely a result of equilibrium fractionation during progressive cooling and Rayleigh condensation of the air mass from the marine moisture source region (Craig, 1961). Dansgaard (1964) studied the variations of δD-δ¹⁸O slopes and found that small slope differences were produced by adiabatic vs. isobaric cooling mechanisms. However, he considered that the amount of deviation from a line of slope 8 for any given cooling trajectory is sufficiently small, such that

any departure of a pair of δD and δ¹⁸O ratios from the MWL can be attributed to kinetic fractionation during evaporation under conditions that deviate from the global mean evaporative conditions. However, on smaller spatial or temporal scales, these smaller deviations prove to be significant where the slope of the relationship between δD and δ¹⁸O values can depart from the global value of 8. In what follows, we explore several important factors that control the slope of δD-δ¹⁸O lines.

As discussed earlier, one primary mechanism that alters the slope of the δD-δ¹⁸O line is changes in moisture source

conditions. For the purpose of the work in this study, we will focus on how seasonal scale moisture source changes affect the slope, but the same processes can be observed over any number of timescales. At most mid- to high-latitude locations in the northern hemisphere, the annual slope of precipitation δD-δ¹⁸O lines is less than eight. Feng et al. (2009) explained such observations as a result of the summer northward migration of the subtropical moisture source to areas with cooler sea surface temperatures and higher relative humidity. Consequently, d-excess has an anti-phase relationship with δ¹⁸O, where d-

excess reaches a maximum in winter and a minimum in summer. In addition to marine sources, there can be other significant sources of moisture, including evaporation from lakes, evapotranspiration from land surfaces, or sublimation from snow surfaces. As an example pertinent to this study, Kopec et al. (2019) suggest that sublimated water vapor contributes significantly to summer precipitation at Summit, Greenland. This moisture has a high d-excess, and thus the annual cycle of d-excess at Summit has a maximum in the summer and a minimum in winter. This d-excess annual cycle, in theory,

corresponds to a δD-δ¹⁸O line with a slope greater than 8. Seasonal variations in the contributions from any of these different sources would determine whether slopes increase or decrease from 8. Likewise, in this work, the deviation of the slope from eight is interpreted, in part, to reflect seasonal variations in the moisture sources.

Another cause of δD-δ¹⁸O slope changes is the process of Rayleigh distillation. While the Rayleigh process is often assumed to change δD and δ¹⁸O values with a constant slope, the distillation trajectory over a large range of isotope ratios

does not have a constant slope (Jouzel and Merlivat, 1984, Ciais and Jouzel, 1994). Dutsch et al. (2017) discuss how simple equilibrium Rayleigh distillation produces nonlinear trajectories for the δD vs. δ¹⁸O relationship, with increasing slope at the earlier stage of condensation but decreasing slope at the later stage. This nonlinear evolution is a result of two competing mechanisms: 1) that the equilibrium fractionation factor increases upon cooling more pronouncedly for hydrogen than for



oxygen isotopes; and 2) that the isotopic ratio of vapor (or liquid) is a nonlinear function of $F$, the fraction of vapor remaining in the air mass. The effect of the former mechanism is to increase the slope of the δD-δ¹⁸O line as temperature decreases, while the effect of the latter mechanism is to decrease the slope of the δD-δ¹⁸O line with increasing rainout of the air mass or decreasing $F$. Consequently, the combined effect produced by the Rayleigh process systematically changes, first

increasing the slope of the δD-δ¹⁸O line and then decreasing the slope. Such a nonlinear feature in the δD-δ¹⁸O distillation trajectory has been both modelled (e.g. Jouzel and Merlivat, 1984; Ciais and Jouzel, 1994) and observed in surface snow in Antarctica (Petit et al., 1991; Masson-Delmotte et al., 2008), and recently observed in precipitation at Summit, Greenland (Kopec et al. 2019). The reduction of the δD-δ¹⁸O slope in the GRIP record during the glacial period, as identified by Jouzel et al. (2007), is likely due to greater Rayleigh distillation resulting, in part, from the much colder site temperatures during

this time and the larger temperature gradient from the subtropical moisture source region to the precipitation site. The non-linearity of this process has also been recognized and used to examine d-excess changes in Antarctic ice cores (Uemura et al., 2004, 2012; Markle et al., 2017). Although this mechanism has been discussed in some studies, it is not yet fully appreciated for precipitation and ice core interpretations.

         A third factor that can potentially alter seasonal and annual δD-δ¹⁸O slopes away from 8 is the manner in which d-

excess can be changed further by kinetic fractionation in mixed-phase clouds (Jouzel and Merlivat, 1984; Ciais and Jouzel, 1994). Ciais and Jouzel (1994) examined the effects of snow formation on the isotopic composition of precipitation when liquid and solid phases of water both exist in a cloud. In this situation, water vapor is supersaturated with respect to ice but undersaturated with respect to liquid water (Wegener, 1911; Bergeron, 1935; Findeisen, 1938). Evaporation of the liquid phase can occur simultaneously with deposition of vapor onto solid ice particles. This process involves vapor diffusion, thus

causing kinetic fractionation and potentially changing the d-excess in precipitation. Ciais and Jouzel (1994) demonstrate that d-excess non-linearly increases by this process, down to –40°C when only solid ice particles exist. The d-excess can change up to 2‰ (Jouzel and Merlivat, 1984; Ciais and Jouzel, 1994), which potentially changes the slope of the δD-δ¹⁸O line. However, the impact of this effect on d-excess appears to be relatively minor on the annual scale, as suggested by Kopec et al. (2019), and thus annual slope variations because of this effect would also be relatively minor. Therefore, we do not

consider this effect in this work. Nevertheless, there are certainly other situations that call for investigating the effect of this process more closely, such as when examining intra-seasonal variations of slope.

         In summary, previous work shows that the slope of a δD-δ¹⁸O line is a result of several factors, including seasonal changes in moisture sources, the degree of distillation of vapor traveling from the prevailing source to the precipitation site, mixed phase cloud processes, and the cooling history of the air mass. All factors have to be weighed in order to extract

environmental information from a δD-δ¹⁸O slope.

## 4.2 Summit precipitation δD-δ¹⁸O slopes

         For the isotopic range of Summit precipitation (highly distilled and relatively depleted in deuterium and ¹⁸O), changes in the degree of distillation should generally follow the late stage of a Rayleigh trajectory, i.e., along the section of





the δD-δ¹⁸O curve that decreases in slope with increasing deuterium and ¹⁸O depletion. We attribute the low values of seasonal slopes for the summer (7.71) and winter (7.77) observed in the precipitation samples to this mechanism.

Kopec et al. (2019) argued and justified that sublimated vapor is a significant moisture source for the summer precipitation of Summit, Greenland. This mechanism explains the offset between low winter and high summer intercepts (or d-excess) of the δD-δ¹⁸O lines. Sublimation from the snow surface has been shown to reduce the d-excess of the remaining snow, while the vapor removed by sublimation has a high d-excess (Moser and Stichler, 1974; Stichler et al., 2001). We assume that sublimation sourced moisture is insignificant in the winter months (Cullen et al., 2014), and thus the slope of the winter line, as state above, is the expected result of marine sourced moisture that has experienced significant Rayleigh distillation where the slope is less than 8. If there were no sublimation contribution in the summer months, the summer δD-δ¹⁸O line would follow a similar trajectory, but with a lower intercept (or d-excess) than the winter line due to summer marine moisture sourced from locations with cooler sea surfaces and/or higher relative humidity (Feng et al., 2009). However, if during the summer months there is a significant contribution of sublimation sourced moisture, which has a high d-excess, the summer δD-δ¹⁸O line could have a higher intercept than the winter line. If the summer mixture of marine and sublimation sourced moisture precipitates following a typical trajectory of Rayleigh distillation, the summer δD-δ¹⁸O line should exhibit a similar slope to that of the winter δD-δ¹⁸O line. Our observations of seasonal δD-δ¹⁸O slopes (7.71 in summer and 7.77 in winter) and intercepts (7.30 in summer and −6.28 in winter) are consistent with this interpretation (Fig. 1).

Despite the summer and winter δD-δ¹⁸O lines having slopes less than 8, the annual slope of the δD-δ¹⁸O relationship is greater than either of the seasonal slopes (Fig. 1) as a result of high summer d-excess values. In the years of our precipitation collection, this winter-summer separation caused the slope of the annual δD-δ¹⁸O line to be greater than eight (8.27). If we accept this mechanism causing an annual slope to be greater than seasonal slopes, then we can immediately propose two meteorological factors that could affect the observed interannual variations in annual slopes.

The first factor is the relative contribution of sublimated moisture to summer precipitation. The amount of sublimation dictates how high the mean d-excess is for the summer isotopic ratio population. If there were no sublimation in the summer, the vapor would be largely marine sourced and its distillation would likely follow a Rayleigh curve similar to the winter one. This scenario would produce the lowest annual slope, similar to the slope of the seasonal lines. Adding sublimation-sourced moisture forces the summer line to be raised above the winter line and the annual slope to increase. Other things being equal, the greater the proportion of sublimation, the greater the seasonal separation, and thus the higher the slope of the annual δD-δ¹⁸O line.

The second factor is the seasonal distribution of annual precipitation. Because the annual δD-δ¹⁸O line is the best fit to all precipitation observations, the number of data points in the summer vs. winter season determines the weight of each season to the annual line. In an extreme case, for instance, if there were no winter precipitation, the annual line would be the same as the summer line and thus would have a low slope. If the precipitation is evenly distributed (or the same number of





data points) between summer and winter, the two seasons are evenly weighted in the regression, resulting in a higher slope of the δD-δ¹⁸O line if the summer and winter lines are separated.

In addition, sublimation proportion and precipitation seasonal distribution significantly impact one another. We note that without sublimation causing the separation between the two lines, the seasonal distribution of precipitation would have

little impact because the seasonal slopes would be similar. When sublimation increases the d-excess of summer precipitation to cause separation, the precipitation distribution becomes increasingly important with greater separation.

We demonstrate the effects of these factors more rigorously using Monte Carlo simulations, and Fig. 4 can be viewed as one realization of many such simulations. Simulations were conducted for each of the following four scenarios – even seasonal distribution/low sublimation (EL), even distribution/high sublimation (EH), uneven distribution/low

sublimation (UL), and uneven distribution/high sublimation (UH). Each scenario was simulated 100 times, and each time the slope was calculated with 24 pairs of δD and δ¹⁸O values. The even distribution scenario consisted of 12 summer and 12 winter values. The uneven distribution scenario used 16 summer and 8 winter values. We randomly generated δ¹⁸O for each simulation based on the Mersenne-Twister technique (Matsumoto and Nishimura, 1998) as defined by the mean and standard deviation of the precipitation dataset for the summer or winter months (-29.0 ± 6.1‰ and -40.1 ± 5.0‰, respectively). Using

the summer or winter δD-δ¹⁸O regression lines (δD-δ¹⁸O slopes are 7.71 and 7.77, respectively, while the intercept will be specified below), we calculate the δD value from the randomly generated δ¹⁸O value and then add a random error based on the standard deviation of the residuals of the δD value around the seasonal regression line (4.3‰ and 4.7‰, for summer and winter respectively). The separation caused by sublimation is defined by the intercept values (or the difference between the summer and winter lines). The average difference between summer and winter lines is 13.6‰, so we define the low

sublimation scenario intercept difference as 9.6‰ and the high sublimation scenario intercept difference as 17.6‰.

Annual and seasonal slopes for all 100 simulations were calculated, and their means were found, for each scenario. Representative examples of each scenario are shown in Fig. 4. On average, the slope of each scenario yielded the hypothesized results: the EH scenario had the greatest mean slope (mean ± standard error: 8.62 ± 0.02), followed by the UH scenario (8.34 ± 0.02), the EL scenario (8.32 ± 0.02), and the lowest slope produced by the UL scenario (8.09 ± 0.02). Each simulated slope

is significantly different from one another ($p < 0.0001$), except for the simulated slopes for the UH and EL scenarios ($p = 0.48$).

### 4.3 Owen ice core δD-δ¹⁸O slopes

Based on the arguments presented in 4.2, we examine how sublimation proportion and precipitation distribution affect the observed annual δD-δ¹⁸O slopes in the Owen ice core. In this section, we consider only the δD-δ¹⁸O slopes of the Owen core after the effect of post-depositional diffusion is corrected. To quantify the two factors, we define two indices, the

Sublimation Proportion Index (SPI) and the Precipitation Distribution Index (PDI), and then determine how Owen ice core δD-δ¹⁸O slopes are statistically related to the indices.



### 4.3.1 Sublimation Proportion Index

The Sublimation Proportion Index (SPI) quantifies how much sublimated vapor contributes to the total summer Summit precipitation, relative to marine-sourced moisture. The SPI weighs the importance of sublimated moisture in Greenland relative to the total precipitable water vapor above Summit. For the former, we used the annual sublimation (km$^3$

water equivalent/yr or km$^3$ w.e./yr) observed by Box et al. (2006) over the period 1988 to 2004, which overlaps a considerable part of our data set. The annual sublimation is normalized by the area of Greenland (1.71 x 10$^6$ km$^2$) to yield Water Vapor Flux (WVF, km/yr). The values from Box et al. (2006) are total annual sublimation, but we assume that they are strongly proportional to the summer sublimation since most sublimation occurs in the summer (Cullen et al., 2014). The total precipitable water (PW; kg/m$^2$) is extracted from ERA-Interim reanalysis (Dee et al., 2011) for the Summit grid cell

(72-73°N, 38-39°W) and is averaged over the summer months (June, July, and August or JJA). The SPI for a given year (*j*), is the ratio of WVF to PW for that year after each is normalized by its respective mean of all available years (Eq. 3)

$$SPI_j = \frac{\frac{WVF_j}{WVF_{mean}}}{\frac{PW_j}{PW_{mean}}}$$  (3)

High (low) values of SPI correspond to greater (smaller) relative contributions of sublimated vapor to the total summer Summit precipitable water. In using Eq. 3 to examine the precipitation isotope ratios, we assume that the contribution of sublimated vapor to the precipitation is proportional to the SPI (i.e. precipitation is proportional to precipitable vapor).

### 4.3.2 Precipitation Distribution Index

The Precipitation Distribution Index (PDI) quantifies how much the summer precipitation for a given year deviates

from what would occur if precipitation is evenly distributed throughout the year (25% in each season). In the earlier conceptual discussions (Section 4.2), we implied that spring and fall storms do not significantly affect the seasonal and annual analyses. Here, we define PDI as the deviation of summer precipitation relative to all other seasons in order to make it most useful in studying sublimation-sourced moisture, which has its greatest presence in the summer. Using 25% for the reference value in the definition of the PDI (Eq. 4) is justified by the observation that the precipitation at Summit occurs

relatively evenly throughout the year (Dibb and Fahnestock, 2004).

Unfortunately, the precipitation records at Summit do not extend over the period of this study, so we again use the ERA-Interim reanalysis product to estimate precipitation amount (Dee et al., 2011). The ERA-Interim reanalysis precipitation amount for the Summit grid cell was extracted for the summer months (JJA) and annually from 1988 to 2004 to overlap the sublimation record. To calculate the Precipitation Distribution Index (PDI), we take the absolute difference from

0.25 of the summer (JJA P) to annual precipitation (P) fraction





$$PDI_j = \left| 0.25 - \frac{JJA\,P_j}{Annual\,P_j} \right| \qquad (4)$$

A PDI value of zero represents an even distribution (with regards to summer precipitation), and the larger the value, the more uneven the precipitation amount with the same effect when summer precipitation is either high or low relative to 25% of the

annual precipitation.

### 4.3.3 Role of SPI and PDI in Owen ice core slopes

A multiple regression of the diffusion corrected slopes (Fig. 5) against the SPI, PDI, and the cross product of the two, SPI*PDI, yield a significant result (p = 0.0058) with the overall $r^2 = 0.61$. The coefficient of each explanatory variable is significant, with SPI (p = 0.014), PDI (p = 0.0014), and SPI*PDI (p = 0.013) accounting for 14.9, 30.5, and 15.6% of the

total variance, respectively. The resulting expression for the annual slope from the multiple regression is

$$Slope = 2.70\,SPI - 8.71\,PDI - 85.93[(SPI - 1.00)(PDI - 0.07)] + 6.04 \qquad (5)$$

where the values in parentheses are the mean values of SPI (1.00) and PDI (0.07) of our dataset.

Each coefficient in Eq. 5 has the sign consistent with that of the hypothesized relationship. For this particular data set and temporarily ignoring the cross term, the slope increases with SPI (positive coefficient) and decreases with PDI (negative partial coefficient) as SPI and PDI deviate from their respective means. The impact of the cross term is more complicated; Fig. 6 shows how variations in SPI and PDI affect the δD-δ$^{18}$O slope. When the change of SPI and PDI has an opposite sign the cross term contributes to a positive slope change. This is particularly obvious when SPI increases under increasingly even

precipitation distribution (high SPI and low PDI at the lower right corner of Fig. 6). All three terms in Eq. 5 yield positive changes in the slope. When SPI and PDI deviate from the mean in the upper left direction, the first and second terms in Eq. 5 would yield a negative change in the slope, while the third term a positive one. These changes would (at least in part) cancel out, resulting to a relatively flat area in Fig. 6. Ultimately, the cross term will out compete the linear terms, corresponding to a slight slope increase at the far upper left corner. This would not make sense given our physical meaning of SPI and PDI as

well as our hypothesis. However, this corner is at the limit of our data range, and the error of Eq. 5 increases as the system moves far beyond the mean of SPI and PDI. The system behavior near the upper left corner of Fig. 6 requires additional observational support. In the following section, we examine some of the implications of (Eq. 5), particularly the sensitivities of the slope to SPI and PDI variations.

### 4.4 Implications

The arguments presented above call for additional considerations for interpreting climatic conditions recorded in the isotopic composition of ice cores. First and foremost, we show that the slopes of δD-δ$^{18}$O lines observed over different



timescales and from various records (i.e. precipitation or ice cores) can be valuable tools to explore hydrological processes through the climatological controls of the isotopic composition of precipitation. Using the slope of this line adds a new method of inquiry as it holds more information, or at least different information, than simply taking the average δD, δ18O, or d-excess over a given time window. In 4.1, we discussed a variety of climatic factors that would drive changes in the slope,

such as seasonal variations of marine moisture sources and the degree of Rayleigh distillation. In the work presented above, we use storm-by-storm precipitation and high resolution ice core measurements to show how annual δD-δ18O slopes are also driven by changes in the proportion of sublimation sourced moisture and the seasonal distribution of precipitation. Similar investigations could be performed using δD-δ18O slope changes for a record at a single location or multiple records across a region to examine a wide range of climatic questions over a variety of temporal scales (e.g. seasonal, annual, decadal, glacial

cycles) and how these variations change spatially.

In order to use δD-δ18O slope measurements in ice cores, it is critical to account for the effects of diffusion, which we show to have significant impacts on the slope. If done properly, the method we developed can be applied to deeper cores and/or at other locations. From the Owen ice core at Summit, Greenland, we are able to analyze 32 full annual cycles by calculating the slope of the δD-δ18O line for each year, and a deeper core would yield more years before diffusion eventually

renders the annual cycle indiscernible. In a region where accumulation rates are greater than Summit, and hence the effect of vapor diffusion is reduced, it is possible that information about the annual cycle would be preserved to a much deeper depth in the core. If diffusion lengths are sufficiently low relative to the annual layer thickness such that the annual cycle is preserved to the depth where vapor diffusion ceases to take place (e.g. at high accumulation locations in Alaska; see Winski et al., 2018), the annual cycle could be measureable for much greater timescales because solid diffusion is about four orders

of magnitude slower than vapor diffusion (Johnsen et al., 2000). At this point, the limit likely becomes measurement resolution, or other physical factors, rather than diffusion. The advancement of continuous flow analysis systems is reducing this limit (e.g. Jones et al., 2017). If such an ideal site exists, factors such as sublimation proportion or the seasonality of precipitation could theoretically be identified throughout the Holocene or even to the transition out of the glacial period.

Another important observation from this work is that, as first stated in Kopec et al. (2019), sublimated moisture is

demonstrated to be a significant component of precipitation at Summit, Greenland. This is particularly important in two ways. First, sublimation as a high d-excess moisture source must be considered when examining variations of d-excess in ice cores. In most studies, an increase in d-excess along an ice core is typically interpreted to indicate warming and/or lower humidity in the marine moisture source area. Our study demonstrates the possibility of an alternative explanation that the increase of d-excess is caused by an increase in the contribution of sublimated vapor, which could be caused by local

climatic changes that increase sublimation rates, such as an increase of temperature, decrease of relative humidity, and/or an increase in wind speed. Examining the slope of the δD-δ18O line allows for identification of this source and determination of its relative contribution, where, other things being equal, steeper slopes correspond to more sublimation relative to marine sourced moisture. Second, while the precise mass balance computations are beyond the scope of this study, the fact that this





moisture source significantly contributes to summer precipitation shows that moisture recycling is potentially an important component to consider for the mass balance of the Greenland Ice Sheet.

Finally, using the results in 4.3 from the Owen ice core, we present a few sensitivity thought experiments to show how δD-δ[18]O slopes can be used to understand past climatic changes. We examine three climate change scenarios below
using the partial regression coefficients of Eq. 5 (and Fig. 6), two of which are contemporary changes that we could observe with this dataset, and the third of which is a hypothetical scenario under glacial conditions. For the modern scenarios, we use the average conditions observed in the Owen core as the initial conditions for the δD-δ[18]O relationship, prior to an assumed climatic change. The mean values for SPI and PDI over the measurement period (1988-2004) are 1.00 and 0.07, respectively, and this combination of factors corresponds to a δD-δ[18]O slope of 8.11 (Eq. 5).

First, we examine the potential effect of SPI changes on the δD-δ[18]O slope. The Arctic is experiencing rapid changes in climate. One important consequence is the increase of total water vapor in the atmosphere due to warming. The amount of precipitation is also expected to increase as a result of sea ice loss that enhances Arctic evaporation (Bintanja and Selten, 2014; Kopec et al., 2016). With higher precipitable vapor, the SPI is expected to decrease, if sublimation source moisture flux stays the same. Kopec et al. (2016) estimated a 20% increase in Arctic precipitation with a local sea ice extent
reduction of 100,000 km[2] and a constant subtropical moisture contribution. With this sensitivity, if we assume that precipitable water also increases 20% with the sublimation moisture contribution unchanged, the SPI would decrease from the present value of 1.0 to 0.83. Such a SPI change would reduce the δD-δ[18]O slope from 8.11 to 7.69 (shown as scenario 'a' in Fig. 6). In terms of the relationship between d-excess and δ[18]O (or δD), this slope shift transfers the seasonal cycles from being relatively in-phase to relatively out of phase. In reality, sublimation may also change in response to warming. The
actual change in SPI depends on relative changes of both sublimation and total precipitable vapor. However, this simple thought experiment demonstrates that the δD-δ[18]O slope should respond to changes of the moisture sources and water budget for Greenland.

Similarly, increasing precipitation from sea ice loss may also affect the PDI and thus the δD-δ[18]O slope. If the increase of precipitation is spread evenly across all seasons, there will be no change of PDI by this effect; otherwise, the PDI
will change. For example, if the previously cited 20% precipitation increase for a 100,000 km[2] sea ice loss all falls evenly in the summer and fall months (the months with lowest sea ice extents), the percentage of summer precipitation will increase from 32% (the current mean summer precipitation fraction corresponding to the mean PDI = 0.07) to 38%. The correspondent change in PDI is +0.06. At the mean SPI value of 1.0, this PDI change will reduce the δD-δ[18]O slope from 8.11 to 7.69 (Fig. 6, scenario 'b'). Again, d-excess changes from being in-phase to out of phase with δ[18]O.

Over the measurement period of the Owen ice core, the reduction of sea ice has caused an increase of Arctic sourced moisture at Arctic coastal sites (Kopec et al., 2016). If this sea ice effect has also reached Summit, Greenland, we would expect to see the δD-δ[18]O slope decrease over time. However, after correcting for diffusion, there is no significant temporal trend in the δD-δ[18]O slope (Fig. 3). Between 1988 and 2004, the SPI value does show a significant decrease (p = 0.04) of 0.01 per year, corresponding to a calculated reduction of slope of 0.24, assuming the initial conditions are mean





values of SPI and PDI. The PDI, on the other hand, does not exhibit any significant trend (p = 0.32) over that same period. It is entirely possible that over this time window that changing precipitation by the loss of sea ice impacts primarily coastal areas and less so Summit, which would be consistent with the argument presented by Kopec et al. (2019) that other factors aside from these marine moisture sources significantly impact the isotopic composition of precipitation at Summit.

Significant additional analysis, such as examining water vapor back-trajectories, would be needed to best understand how the connection between sea ice and precipitation is recorded at Summit.

Last, we examine a scenario in glacial periods when winter precipitation was thought to have been significantly reduced to near zero (Werner et al., 2000). The initial condition is assumed to have evenly distributed precipitation (PDI = 0). If we take that scenario to its limit where winter precipitation were removed completely, and precipitation in other seasons

remains the same in the relative proportion (evenly distributed in spring, summer and fall), the summer/annual precipitation ratio would increase to 33% (PDI = 0.08). At a constant SPI value of 1.05 (an assumed value for convenient illustration in Fig. 6 scenario 'c'), this change of precipitation distribution would cause a reduction of the $\delta$D-$\delta^{18}$O slope from 9.02 to 8.12. While observations in deep ice cores have not been measured at the resolution necessary to test this hypothesis, the framework we present here provides a new method to explore changes, such as the precipitation seasonality, back in time, whether at Summit

or other locations.

## 5 Conclusions

In this study, we demonstrate that the slope of the $\delta$D-$\delta^{18}$O line contains significant climatic information. We measured the slope of this line for event based precipitation and the shallow Owen ice core from Summit, Greenland. The precipitation measurements yielded an annual slope of the MWL of 8.27 from 2011 to 2014, consistent with the high

summer and low winter d-excess values. On the other hand, isotopic variations within summer and winter seasons considered separately yielded slopes of 7.71 and 7.77, respectively. The annual $\delta$D-$\delta^{18}$O slope of the Owen ice core was computed for each year from 1978 to 2009, and the average annual slope of the raw data was 8.67. The annual amplitude of the isotope cycles were shown to be significantly reduced by diffusion at the lower depths of the ice core and the slope was altered, but we corrected the $\delta$D-$\delta^{18}$O slope with a diffusion model. After the correction, the majority of years still had an

annual slope greater than eight, despite the overall reduction of the average annual slope to 8.04.

The slope of the seasonal MWLs in Summit precipitation is likely controlled by Rayleigh distillation. Under the range of isotope ratios observed at Summit, it is expected that Rayleigh distillation causes the d-excess to increase with greater depletion of $\delta$D or $\delta^{18}$O, thus yielding a slope of the $\delta$D-$\delta^{18}$O line below eight. Although winter and summer slopes are similar, the summer line has a significantly greater intercept (or mean d-excess). Among many known mechanisms that

cause increases in d-excess, sublimation-sourced moisture is most likely responsible for the high d-excess value of the Summit summer precipitation. This work provides additional evidence that significant moisture recycling may occur on the Greenland Ice Sheet and should be considering when calculating glacial mass balance.





The offset between winter and summer isotopic distribution results in a greater slope of the annual MWL compared to its seasonal counterparts. Since the annual $\delta D$-$\delta^{18}O$ line is a weighted combination of the seasonal lines, two primary controls can be hypothesized to cause the annual slope variations 1) the proportion of sublimation-sourced moisture in the total summer precipitation and 2) the seasonal distribution of precipitation amount. The greater the amount of separation

between winter and summer due to summer contribution from sublimation vapor, the higher the annual slope. The weighting of each seasonal line, determined by the intra-annual distribution of precipitation, also impacts the slope; with a given level of sublimation-sourced moisture the more evenly the precipitation is distributed across seasons, the greater the slope. Using a Monte Carlo simulation, we show that both factors significantly control the slope of the annual $\delta D$-$\delta^{18}O$ line.

To test the hypothesis that the proportion of sublimation-sourced moisture and the seasonal distribution of

precipitation amount control the annual slope in ice cores, and to quantify the impact of these two explanatory factors, we defined indices for each: the sublimation proportion index (SPI) and precipitation distribution index (PDI). The SPI is a measure of the relative importance of sublimation moisture contribution to the total summer precipitation. The PDI expresses the degree of evenness in the seasonal precipitation distribution within a year, or more specifically, how summer precipitation amount deviates from 25% of the total yearly precipitation. Both variables and their interaction significantly

predict the diffusion corrected slope of annual $\delta D$-$\delta^{18}O$ line measurements in the Owen ice core. We obtain the expected signs for all of the partial coefficients; increases in SPI (positive changes), decreases in PDI (negative changes), and their respective interactions (SPI*PDI) yield sharpest increases in the slope.

These observations have several implications for ice core interpretations. We show that the slope of the $\delta D$-$\delta^{18}O$ line contains significant environmental information at Summit and thus can be used as a paleoclimate tool for other ice cores.

Deeper cores at Summit, if seasonally resolved, could potentially reveal the annual cycle further back in time, and thus allow for the reconstruction of changes of sublimation-sourced moisture and/or precipitation seasonality. At a hypothetical site with a higher accumulation rate than Summit, the annual cycle could be observed over much greater timescales and these factors, or others that control the slope, could be examined. We also examine the sensitivity of SPI and PDI to changes of the hydrologic cycle, and predict that an increase of Arctic sourced moisture from loss of sea ice could significantly change the

slope of the annual $\delta D$-$\delta^{18}O$ line.

While water isotopic variations have been widely used in many disciplines, few studies have explored variations in slopes of the annual $\delta D$-$\delta^{18}O$ lines over time. This study has contributed new methodology for such an analysis, and arrived at important understanding of systematic variations in water isotopic ratios. In terms of paleoclimate reconstructions using precipitation-derived isotopic proxies, this work provides new perspective and the groundwork for future innovative uses of

the water isotope system, as well as alternative interpretations of isotopic variations in ice cores (e.g., d-excess).



**Data availability**

Precipitation collection data and respective water isotope measurements is available in Kopec et al. (2019). Owen ice core data is available in the NSF Arctic Data Center (https://doi.org/10.18739/A21J9774C).

**Author contribution**

1. Conceptualization: BGK, XF, ECO, ESP

2. Data curation: BGK, XF, ECO

3. Formal analysis: BGK, XF, ECO, ESP

4. Funding acquisition: XF, ECO, ESP

5. Investigation: BGK, XF, ECO, ESP

6. Methodology: BGK, XF, ECO, ESP

7. Project administration: XF, ECO, ESP

8. Resources: BGK, XF, ECO, ESP

9. Software: N/A

10. Supervision: XF, ECO, ESP

11. Validation: BGK, XF, ECO, ESP

12. Visualization: BGK, XF, ECO, ESP

13. Writing – original draft: BGK, XF

14. Writing – review & editing: BGK, XF, ECO, ESP

**Acknowledgements**

This work was supported by the National Science Foundation under grants 1022032 for the iisPACS (Isotopic Investigation of Sea ice and Precipitation in the Arctic Climate System) project and award ARC-0909265 to Robert Hawley. The project was made possible by the dedicated work of the Polar Field Services science technicians to maintain the sampling system at Summit and to collect precipitation samples. We thank Robert Hawley, Gifford Wong, Zoe Courville, and Elle Anderson for their role in the collection and processing of the Owen ice core, and Gifford Wong, Jenny Howley, and David Ferris for its

isotopic analysis. We thank Mike Handley at the University of Maine for analyzing the core for elemental concentrations with an ICPMS. Logistical support from the Ice Drilling Programs Office, 109[th] Air National Guard, Polar Field Services is greatly appreciated. We thank Leslie Sonder, Meredith Kelly, and James White for their insightful contribution to discussions and improvement of this manuscript.



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



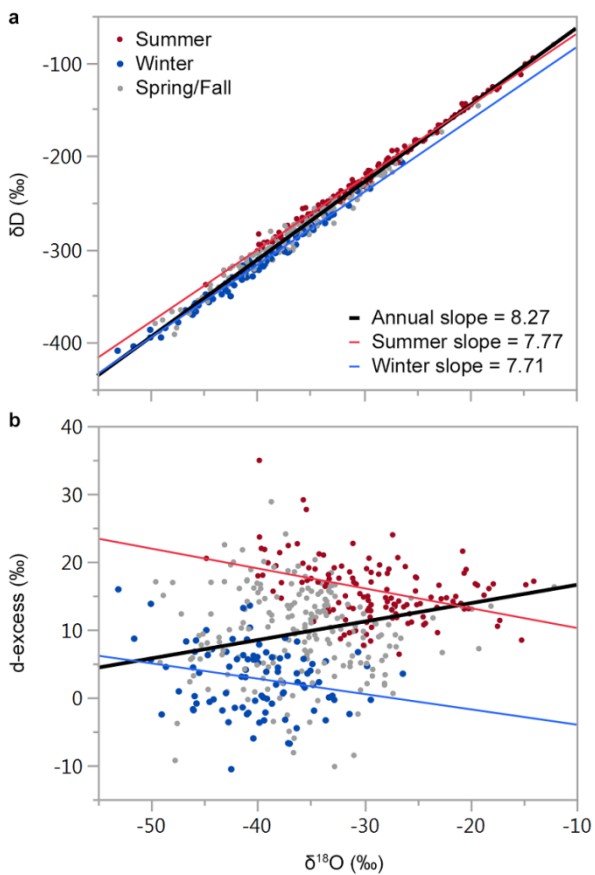

**Figure 1:** (a) Plot of δD vs. δ¹⁸O for entire precipitation dataset broken down seasonally into summer (red), winter (blue), and spring and fall (gray) points. The annual regression line is shown (black line) in addition to seasonal regression lines for summer (red line) and winter (blue line), respectively. (b) Plot of d-excess vs. δ¹⁸O with the same points and lines as in (a).

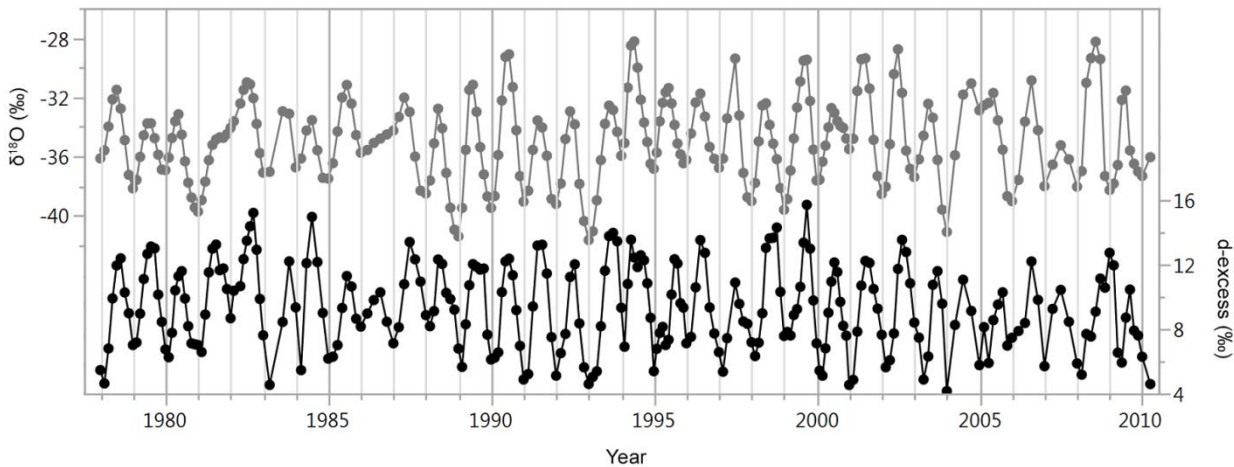

**Figure 2:** Raw measurements of δ¹⁸O (gray points and line) and d-excess (black points and line) over the length of the Owen ice core. Data extends from 1978 to early 2010.





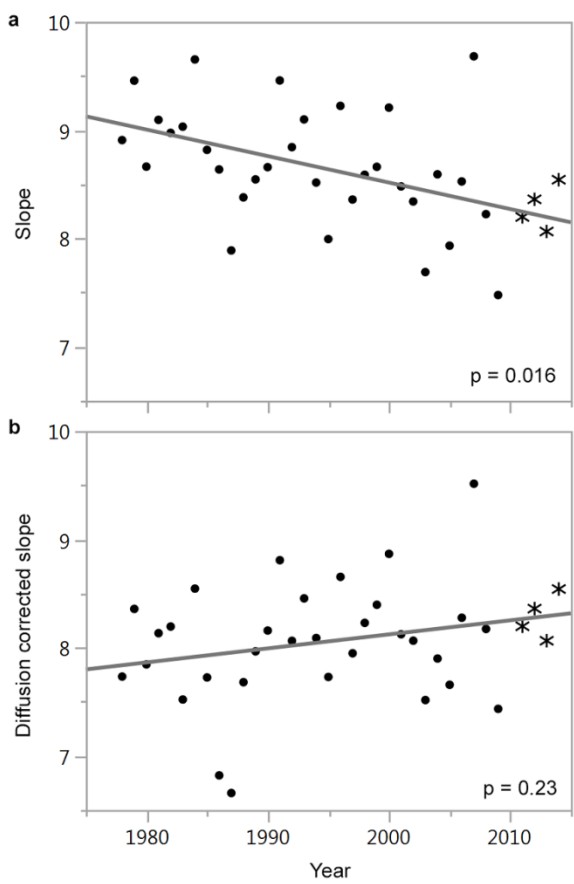

**Figure 3:** (a) Annual δD-δ18O slopes calculated from raw data for the Owen ice core. Regression of slope vs. year is significant ($r^2 = 0.18$, $p = 0.016$). (b) Diffusion corrected annual δD-δ$^{18}$O slopes as defined in section 2.4. Regression of slope vs. year is not significant ($r^2 = 0.05$, $p = 0.23$). The slope of δD-δ$^{18}$O lines for precipitation samples for years from 2011 to 2014 are shown as (*), but not included in the
5   regression analysis.



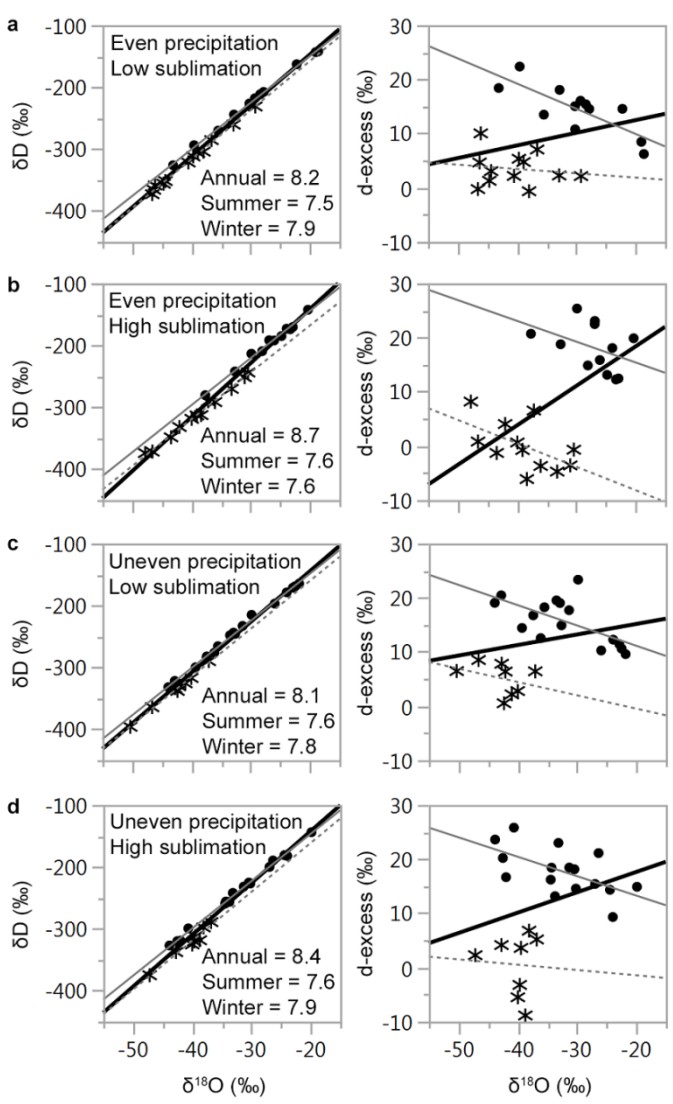

**Figure 4:** One example of Monte Carlo simulations of δD vs. δ¹⁸O (left) and d-excess vs. δ¹⁸O (right) relationships for scenarios labeled in the plots of the left column. Simulated summer (●) and winter (*) values are shown. The regression lines are solid black for annual, solid gray for summer, and dashed gray for winter relationships. See the text for a more detailed explanation.


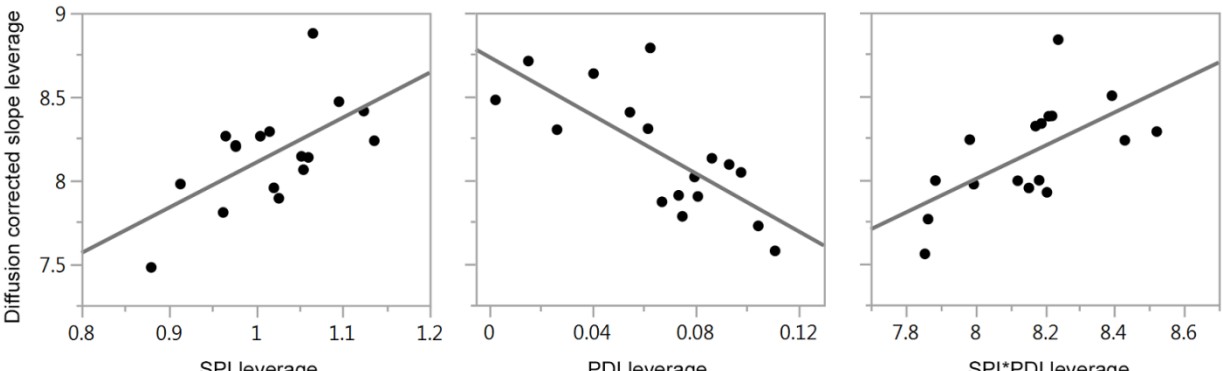

**Figure 5:** Leverage plots (Sall, 1990) for the multiple regression of diffusion corrected annual $\delta D$-$\delta^{18}O$ slopes against Sublimation Proportion Index (SPI; $p = 0.014$), Precipitation Distribution Index (PDI; $p = 0.0014$), and the interaction between SPI and PDI ($p = 0.013$). The regression yields an overall $r^2 = 0.61$, $p = 0.0058$, and root mean square error = 0.27.

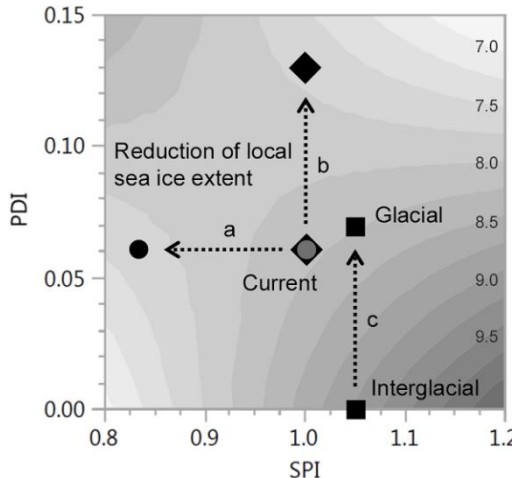

**Figure 6:** Contour plot showing $\delta D$-$\delta^{18}O$ slopes as a function of PDI and SPI. The current conditions are displayed where SPI = 1.0 and PDI = 0.07, the values averaged over the years 1988-2004. This corresponds to a $\delta D$-$\delta^{18}O$ slope of 8.11. Three scenarios, labeled a, b and c, of climatic changes are displayed. a) A change of slope caused by a reduction in SPI due to an increase in marine sourced moisture (displayed as ●). b) A change of slope caused by an increase in PDI due to an increase in the summer and fall precipitation (displayed as ◆). c) A change

10 of slope expected by a change of PDI due to cut off of winter precipitation, an assumed extreme condition during glacial times (displayed as ■). See the text for detailed explanations of these scenarios.