# Peer review of "Precipitation and ice core $\delta D - \delta^{18}O$ line slopes and their climatological significance"

_Climate of the Past, 2019_

## Referee Comment (RC1) · Anonymous Referee #1 · 28 Jul 2019

The manuscript submitted by Kopec and others deals with the relationship between d18O and dD in Greenland with new data from the Owen ice core drilled at Summit and covering the period 1977 to 2010. The main message covey by this manuscript is a discussion of the slope between dD and d18O with difference in summer and winter that is attributed to an important contribution of surface sublimation in Greenland to the precipitation at Summit in summer. The authors also propose a way to link the slope to a budget of sublimation vs precipitation amount.

I can not support the publication of such manuscript for many reasons given below: - The authors can not ignore all the recent literature on the d18O – d-excess in surface snow and shallow firn cores showing different results that those presented here. As an example, Steen-Larsen et al. (2011) did a very detailed analysis of d18O – d-excess

variations on a shallow ice core at NEEM. This study was followed by the manuscript of Masson-Delmotte, Steen-Larsen et al. (2015) with much more data. In these two manuscripts, the link between d18O and d-excess is clearly different from what is presented in the present paper. The interpretation is thus different as well, with an important contribution of marine source evaporation in the whole d-excess signal. I don't challenge the measurements performed in the present study since the water isotopic measurements are routine work but it is not correct to present new data contradicting previous recent ones in ignoring this work. This is particularly problematic since the authors propose an interpretation by using the global sublimation flux over Greenland and not any regional estimate (or calculated using backtrajectories for example) so that there is no reason why the explanation proposed in the present study should not be valid for another Greenland site. - In the same line, Steen-Larsen, Bonne and others (see some references at the end of the review) have largely studied the imprint of evaporation over the ocean and sublimation over the snow in Greenland on the d18O, dD and hence d-excess signals both with monitoring of the water vapor isotopic composition and with modelling approaches including water isotopes. Again, the authors does not quote any of these studies and only quote for sublimation some older papers that are even not listed in the reference list (Moser and Stichler, 1974; Stichler et al., 2001). - In addition to the recent literature, older papers are also fully ignored such as the study of Hoffmann et al. (2001) presenting a fully different interpretation of the recent d-excess signal at GRIP, i.e. at Summit, and a different signal too. - The dating of the Owen ice core is not described sufficiently while the whole analysis is dependent on this dating. A whole section should be devoted to this aspect showing the chemical concentrations, how they are used to date the ice. In the present manuscript, this section is not robust enough to support the conclusion. - I am very concerned about the way diffusion is treated in this paper. Even if we consider the simple diffusion model of Johnsen correct, it is not used in the right way here. Indeed, in the initial paper by Johnsen et al. (2000), it is stated on section 2.2.3 (p. 171) that an artificial signal of d-excess is created by diffusion and this is observed in the figure 4 of this paper. In

other words, the diffusion does not only play a role in the amplitude of the d-excess signal as mentioned and calculated in this study but also on the phasing between the d18O and d-excess signals. Steen-Larsen et al. (2013) used the Johnsen model and corrected then for a phasing between d-excess and d18O on the NEEM shallow ice core. I am very surprised that the authors do fully ignore this effect which probably fully biases their analysis and interpretation. I am also very surprised that the authors only show the raw series of d18O and d-excess and never the diffusion corrected series.

Summarizing, I have serious doubts on the robustness of the dating and diffusion correction of the series presented here to follow the interpretation proposed. Moreover, the ignorance of a rich and documented literature on the subject limits the scientific interest of the present study for a large community working on water isotopes in the high latitudes of the northern hemisphere.

Bonne, J. L., Behrens, M., Meyer, H., Kipfstuhl, S., Rabe, B., Schönicke, L., ... & Werner, M. (2019). Resolving the controls of water vapour isotopes in the Atlantic sector. Nature communications, 10(1), 1632. Hoffmann, G., Jouzel, J., & Johnsen, S. (2001). Deuterium excess record from central Greenland over the last millennium: Hints of a North Atlantic signal during the Little Ice Age. Journal of Geophysical Research: Atmospheres, 106(D13), 14265-14274 Johnsen, S. J., Clausen, H. B., Cuffey, K. M., Hoffmann, G., Schwander, J., & Creyts, T. (2000). Diffusion of stable isotopes in polar firn and ice: the isotope effect in firn diffusion. In Physics of ice core records (pp. 121-140). Hokkaido University Press. Madsen, M. V., Steen‐Larsen, H. C., Hörhold, M., Box, J., Berben, S. M. P., Capron, E., ... & Kipfstuhl, S. (2019). Evidence of isotopic fractionation during vapor exchange between the atmosphere and the snow surface in Greenland. Journal of Geophysical Research: Atmospheres, 124(6), 2932-2945. Masson-Delmotte, V., Steen-Larsen, H. C., Ortega, P., Swingedouw, D., Popp, T., Vinther, B. M., ... & Falourd, S. (2015). Recent changes in north-west Greenland climate documented by NEEM shallow ice core data and simulations, and implications for past-temperature reconstructions. Steen‐Larsen, H. C., Masson‐Delmotte,

V., Sjolte, J., Johnsen, S. J., Vinther, B. M., Bréon, F. M., ... & Gallée, H. (2011). Understanding the climatic signal in the water stable isotope records from the NEEM shallow firn/ice cores in northwest Greenland. Journal of Geophysical Research: Atmospheres, 116(D6) Steen-Larsen, H. C., Masson-Delmotte, V., Hirabayashi, M., Winkler, R., Satow, K., Prié, F., ... & Dumont, M. (2014). What controls the isotopic composition of Greenland surface snow?. Climate of the Past, 10(1), 377-392 Steen‐Larsen, H. C., Risi, C., Werner, M., Yoshimura, K., & Masson‐Delmotte, V. (2017). Evaluating the skills of isotope‐enabled general circulation models against in situ atmospheric water vapor isotope observations. Journal of Geophysical Research: Atmospheres, 122(1), 246-263.

---

## Referee Comment (RC2) · Mathieu Casado (Referee) · 25 Aug 2019

**Review of « Precipitation and ice core δD-δ18O line slopes and their climatological significance» by Ben G. Kopec, et al.**

This paper presents new measurements of water isotopic composition from the site of Summit in Greenland where several ice cores have been drilled. The results compare precipitation isotopic composition to an ice core record and highlight a peculiar relationship between $\delta^{18}O$ and $\delta D$ (greater than 8), leading to values of d-excess singularly lower to what is usually found in Greenland. They interpret this different behaviour to the impact of sublimation on the snow isotopic composition, and thus suggest that a large part of the d-excess signal represent the sublimation over the ice sheet, and not the evaporation conditions as commonly interpreted.

I read the manuscript with great interests, it rises an interesting and plausible alternative explanation of the slope between $\delta^{18}O$ and $\delta D$ affecting the interpretation of the isotopic paleothermometer. However, I have several concerns of major and minor nature, which address a few methodological aspects as well as the description of the relevant processes.

General comments:

1. In this manuscript, the authors suggest that the slope between $\delta^{18}O$ and $\delta D$ found at summit is an anomaly basing their studies on the results of Feng et al, 2009. The new results from summit seems to present positively correlated variations of d-excess against $\delta^{18}O$ while GNIP data (Feng et al, 2009) indicate that the normal behaviour is negatively correlated at the seasonal scale.

   The negatively correlated relation between d-excess and $\delta^{18}O$ is normal for low isotopic composition areas due to the Rayleigh distillation. Typically, for a Rayleigh distillation process, the local variations of $\delta^{18}O$ and $\delta D$ can be calculated by:

   $$\frac{d\delta D}{d\delta^{18}O} = \frac{\alpha_D - 1}{\alpha_{18} - 1} \frac{1 + \delta D}{1 + \delta^{18}O}$$

   where $\alpha_D$ and $\alpha_{18}$ are the respective equilibrium fractionation coefficients (or effective if you include kinetic fractionation). For very low temperature conditions, $\frac{\alpha_D - 1}{\alpha_{18} - 1} \approx 10$ (typically for -30°C, you would have 9.65). Yet, for low isotopic composition such as found at summit ($\delta D \approx -300‰$, $\delta^{18}O \approx -35‰$), This means the expected slope between $\delta^{18}O$ and $\delta D$ would be 7‰/‰. This effect has been described in multiple papers such as (Touzeau et al., 2016;Casado et al., 2016).
   This generalise to any time scales the results of Feng et al (2009), and thus is key for your manuscript where you compare precipitation isotopic composition at the seasonal scale to an ice core records spanning more than 30 years. It also shows that in both winter and summer, the slopes you obtained are larger than expected by the Rayleigh distillation.

2. There are previous observations of positively correlated d-excess and $\delta^{18}O$ (or $\delta D$) in Greenland. For instance, Barlow et al. (1993) present data from Summit where $\delta D$ and d-excess are correlated at the seasonal scale in an ice core. This seesaw variation was later attributed before to variations of SST in the north atlantic (Hoffmann et al., 2001) using a Rayleigh distillation model. In depth discussion on the difference between your results and their results are necessary.

3. Section 4.1 is essentially a second introduction. I recommend that you to consider integrating it into the introduction and removing it from here.

   Oppositely, what is missing from the discussion is the comparison of your results with time series and slopes from other study at summit and in other neighbouring sites in Greenland.

4. Section 4.2 is essentially a method sub-section, except for the first two paragraphs which are introduction material. Consider integrate it into the methods and removing it from here.

5. The arguments in section 4.3, which are central to the arguments of this manuscript, are developed with unsatisfactory level of details.

   First, it is not clear what data are used to calibrate the multi-linear regression problem in section 4.3.3. and in Fig. 5. Only 16 diffusion corrected slope data points appear on the plots while you initially have 32 years obtained from the owen ice core. Were the data selected to cover the period observed in Box et al, 2006 ? If yes, is this period representative of the entire period ?

   Second, is the SPI over the entire surface of Greenland representative of the SPI across the area where moisture travels from the North Atlantic to Summit ? As there is a significant surface north of Summit where one does not expect any summer precipitation travel path to go to, it would be important to evaluate how this impacts the data. Are the precipitation also coming more generally from trajectories from the Davis Straight of the Denmark Straight ? Considering that the patterns observed are very different for the South East, the South West, and the North, one would expect this to impact the results. In general, changes of origin of the air masses from the west or the east of Greenland would be expected to impact the results, and is not discussed here relatively to the impact of sublimation.

   Third, how good is ERA-interim precipitable water amount ? In general, the amount of precipitation over the ice sheets is quite biased.

   There is no interpretation of the various regressions in Fig. 5. For instance, earlier in the manuscript, you describe the temporal slope of the diffusion corrected slopes as not significant. So how are the trends compared to the original trend ?
   It seems also that the vertical axes of the three different sub-figures of figure 5 are different as the horizontal levels of the larger and smaller vertical values do not match (I can't judge for the other points).
   How crucial is the diffusion correction for these results ?
   How good would be a prediction from the multi-linear regression using the best fit to actually predict your slope time serie ?

6. It is not clear to me how you discard the change of trajectory of the air masses between summer and winter, which could also explain the offset between summer and winter values.

For instance, have you performed any back-trajectory analysis that shows that for the period from 1979 to today, the air masses originates and travel roughly across the same areas and are under the same level of distillation ?

7. As you mention the large impact of sublimation across the Greenland ice sheet to the moisture that leads to the formation of precipitation, I think you need to mention the impact on the surface snow (see the comments of the other reviewers).

8. Alternatively, there are measurements of vapour isotopic composition at Summit (Berkelhammer et al., 2016) which would provide an estimation of the vapour d-excess value which can be of interest to validate the hypothesis that sublimation from the ice sheet provides a significant amount of moisture into the precipitation. The impact of the exchanges between the snow and the vapour could also affect the surface d-excess.

9. The study presented here focuses mainly on seasonal and interannual time scales. The previous study of the authors (Kopec et al, 2019) focuses on the synoptic time scale. In general, I do not believe it is possible to generalise the results from these time scales to large time scale without further studies and recommend caution to the authors to really assess at which time scales they results can be applied.

10. In general, the authors need to include a greater representation of the relevant literature. There are a lot of papers that would be relevant for this study that have been overlooked, which as a result weaken the manuscript.

Specific comments:

Page 2 – Line 17: "Over the past half century, variations of hydrogen ($\delta D$) and oxygen ($\delta 18O$) isotopic ratios of precipitation have served as increasingly powerful tools in a wide range of disciplines of research, including paleoclimate, hydrology, and atmospheric sciences."

While this is generally true, there is a large body of litterature, including review papers that you can cite here to your point...

Page 2 – Line 19: "One of the most striking features of these two paired isotope ratios is that they are remarkably well correlated over time and space, and this relationship is defined as the meteoric water line (MWL)."

At this stage, one of the early reference could also be used, for instance: (Dansgaard, 1964). You could also add one of the GNIP paper (Schotterer et al., 1996).

Page 2 – line 26: "In the field of paleoclimate studies, for example, $\delta D$ or $\delta 18O$ variations in ice cores have been used to infer temperature changes at ice core drilling sites (e.g. Jouzel et al., 1987; Dansgaard et al., 1989; Johnsen et al., 1995, 2001; Blunier et al., 2001; Petit et al., 1999) and deuterium-excess variations in ice cores have been used to infer

changing marine moisture source conditions over a range of time scales (e.g. Johnsen et al., 1989; Barlow, et al., 1993; Vimeux et al., 1999, 2001; Uemura et al., 2004, 2012; Masson-Delmotte, et al., 2005; Jouzel et al., 2007)."

For temperature reconstruction, there are a large number of recent reference that could be included, including from Greenland: (NEEM, 2013;NorthGRIP, 2004)...

For the d-excess variations and the link with marine moisture source conditions, I would also recommend including more recent references

Page 3 – Line 1: "There are relatively fewer studies focusing on how the slope of the MWL changes over space and time, and determining how such variations may contain climate information."

This sentence needs to be more precise:

1. the slope of the MWL in the isotopic composition of precipitation is what you're focusing on, as the MWL slope can change in the snow for instance due to diffusion, and thus is not containing any climatic information
2. Fewer than what ?
3. I'm not convinced that there are few studies which have looked at the spatial variations of the slope $\delta D$ vs $\delta^{18}O$, for instance (Masson-Delmotte et al., 2008;Touzeau et al., 2016;Landais et al., 2017;Werner et al., 2018;Jouzel et al., 2000), for the temporal variations (Oyabu et al., 2016;Masson-Delmotte et al., 2015;Werner et al., 2001;Steen-Larsen et al., 2011;Persson et al., 2011) and for the space-time (Hendricks et al., 2000;Risi et al., 2013;Sodemann et al., 2008;Ekaykin et al., 2002)

Page 3 – Line 3: "For example, if the slope of a LMWL with seasonally resolved observations is less than 8, then d-excess and δD (or δ18O) would have an anti-phase relationship, and the opposite is also true "

I believe this is generally true, regardless of the seasonal resolution.

Page 3 – Line 6: "This out-of-phase relationship between the two is equivalent to the fact that, in these locations, the LMWL (with a monthly or higher resolution) has a slope less than 8. "

Here, the relationship is in "phase opposition", not out of phase. Out of phase could mean everything as completely random, to in phase opposition.

Page 3 – Line 10: "To our knowledge, this is the only work that described the slope distribution of LMWLs on a hemispheric scale and discussed the climatological significance of these slopes."

This is a very strong statement. For instance, this study (Pfahl and Sodemann, 2014) seems to do something quite similar using d-excess, which is equivalent to the slope δD-δ18O.

Page 3 – Line 11: "Temporal changes, e.g., seasonal or inter-annual, in the δD-δ18O relationship for a given location have not been explored and can potentially provide information and understanding of seasonal or interannual variations in the planet's climate system."

There is body of research exploring for a given location the δD-δ18O relationship. In Antarctica: (Touzeau et al., 2016;Dittmann et al., 2016;Stenni et al., 2016). In Greenland: (Landais et al., 2012;Steen-Larsen et al., 2011)

Page 3 – Line 14: "Summit, Greenland is one of the most important sources of deep ice cores that provide valuable paleoclimate records, particularly through the measurement of water isotopes. "

I think the term "most important source of deep ice cores" is not very precise. Please reformulate.

Page 3 – Line 17: "While Feng et al. (2009) demonstrated that almost all sites in the mid- to high-latitudes of the Northern (and Southern) hemisphere exhibit an out-of-phase relationship between δD and d-excess, and the mechanisms controlling this pattern are relatively well understood, Kopec et al. (2019) recently reported a nearly in phase relationship between δD and d-excess of event-based precipitation measurements at Summit, Greenland."

The demonstration from Feng et al (2009) is at the seasonal scale from GNIP data. δD and d-excess are there anticorrelated.

Your previous study (Kopec et al, 2019) presents results across the synoptic scale and the seasonal scale.

Both results are not necessarily opposed considering the different time scales.

Page 6 – Line 22: "This result is consistent with the observations of Kopec et al. (2019), where they reported dexcess measurements in phase with δD or δ18O values."

Why don't you present the slope and the correlation ? "In phase" is vague to describe the link between d-excess and δD.

Page 7 – section 3.2: As you are comparing results obtain from precipitation samples and from an ice core, it would be important to transpose the diagnosis made in section 3.1 to section 3.2. In particular, I think presenting the equivalent of figure 1.b) for the Owen ice core would be beneficial. In the precipitation, you show that the slopes are actually smaller than 8 if you look at winter and summer separately, but there is a shift between the cloud of winter points and the cloud of summer points. Is the same shift visible in the Owen ice core if you do the slope on all the summer points (high $\delta^{18}O$ points) and all the winter points ?

Page 7 – Line 23: "Correction for the diffusion effect on the isotopic record produces some significant differences, most prominently, the elimination of the temporal trend in the δD-δ18O slope"

Did you realise any sensitivity tests for this ? Considering this is widely used after, it would make sense to make sure that this is pertinent.

Also, considering other ice core have been drilled at summit in the 90's, can you compare the values with these ice cores ? It would be interesting as then, the values at the top of these old ice core would not have had time to be diffused, and thus, you can validate how much the back diffusion is not creating any artefacts.

Page 10 – Line 5: "Sublimation from the snow surface has been shown to reduce the d-excess of the remaining snow, while the vapor removed by sublimation has a high d-excess (Moser and Stichler, 1974; Stichler et al., 2001)."

There are more recent studies studying these processes (Sokratov and Golubev, 2009;Steen-Larsen et al., 2013;Steen-Larsen et al., 2014;Casado et al., 2016;Ritter et al., 2016)

Page 13 – Line 15: As figure 6 does not have a colour scale for the δD-δ18O slope, it is very difficult to understand this paragraph. In general, I would recommend a more detail explanation.

Page 13 – Line 31: "First and foremost, we show that the slopes of δD-δ18O lines observed over different timescales and from various records (i.e. precipitation or ice cores) can be valuable tools to explore hydrological processes through the climatological controls of the isotopic composition of precipitation."

In your study, you present results covering the seasonal and interannual scale. The generalisation to larger time scale is not shown, but hypothesised.

Also, in your manuscript, I was under the belief that you considered ice cores as precipitation (and diffusion which you are correcting). A discussion on the differences between precipitation and ice core would be interesting, but I don't believe it is central to your manuscript.

Page 14 – Line 2: "Using the slope of this line adds a new method of inquiry as it holds more information, or at least different information, than simply taking the average δD, δ18O, or d-excess over a given time window."

d-excess and the slope δD vs δ18O is the same climatic information.

Page 14 – Line 11: "In order to use δD-δ18O slope measurements in ice cores, it is critical to account for the effects of diffusion, which we show to have significant impacts on the slope. If done properly, the method we developed can be applied to deeper cores and/or at other locations."

I also believe this is very important. I suggest you include tests that evaluate what impact the correction of effects of diffusion has on the slopes.

Page 14 – Line 33: "Second, while the precise mass balance computations are beyond the scope of this study, the fact that this moisture source significantly

contributes to summer precipitation shows that moisture recycling is potentially an important component to consider for the mass balance of the Greenland Ice Sheet."

This is indeed out of the scope of the study and brings a lot of questions:

- How much moles of water does your result suggest this represent?
- What is the relative proportion that this represent compared to the summer accumulation ?
- How do you distinguish surface sublimation from sublimation of the snow flakes by katabatic winds in coastal areas (Grazioli et al., 2017) ? Indeed, the sublimation of the later will not contribute to the SMB.

I suggest to remove this sentence or to go in more details.

Page 15 – Line 30: "Over the measurement period of the Owen ice core, the reduction of sea ice has caused an increase of Arctic sourced moisture at Arctic coastal sites (Kopec et al., 2016). If this sea ice effect has also reached Summit, Greenland, we would expect to see the δD-δ18O slope decrease over time. However, after correcting for diffusion, there is no significant temporal trend in the δD-δ18O slope (Fig. 3)."

At this stage, I don't think you can reach such conclusion without evaluating the impact of the diffusion correction on your data and by using a single ice core while multiple previous studies have proposed alternative explanations which fit several ice core, and even sometimes $^{17}O-excess$.

Bibliography:

Barlow, L., White, J., Barry, R., Rogers, J., and Grootes, P.: The North Atlantic oscillation signature in deuterium and deuterium excess signals in the Greenland Ice Sheet Project 2 ice core, 1840–1970, Geophysical Research Letters, 20, 2901-2904, 1993.

Berkelhammer, M., Noone, D. C., Steen-Larsen, H. C., Bailey, A., Cox, C. J., O'Neill, M. S., Schneider, D., Steffen, K., and White, J. W.: Surface-atmosphere decoupling limits accumulation at Summit, Greenland, Science Advances, 2, e1501704, 2016.

Casado, M., Landais, A., Masson-Delmotte, V., Genthon, C., Kerstel, E., Kassi, S., Arnaud, L., Picard, G., Prie, F., Cattani, O., Steen-Larsen, H. C., Vignon, E., and Cermak, P.: Continuous measurements of isotopic composition of water vapour on the East Antarctic Plateau, Atmos. Chem. Phys., 16, 8521-8538, 10.5194/acp-16-8521-2016, 2016.

Dansgaard, W.: Stable isotopes in precipitation, Tellus, 16, 436-468, 10.1111/j.2153-3490.1964.tb00181.x, 1964.

Dittmann, A., Schlosser, E., Masson-Delmotte, V., Powers, J. G., Manning, K. W., Werner, M., and Fujita, K.: Precipitation regime and stable isotopes at Dome Fuji, East Antarctica, Atmospheric Chemistry and Physics, 16, 6883-6900, 2016.

Ekaykin, A. A., Lipenkov, V. Y., Barkov, N. I., Petit, J. R., and Masson-Delmotte, V.: Spatial and temporal variability in isotope composition of recent snow in the vicinity of Vostok station, Antarctica: implications for ice-core record interpretation, Annals of Glaciology, 35, 181-186, 10.3189/172756402781816726, 2002.

Grazioli, J., Madeleine, J.-B., Gallée, H., Forbes, R. M., Genthon, C., Krinner, G., and Berne, A.: Katabatic winds diminish precipitation contribution to the Antarctic ice mass balance, Proceedings of the National Academy of Sciences, 201707633, 2017.

Hendricks, M., DePaolo, D., and Cohen, R.: Space and time variation of δ18O and δD in precipitation: Can paleotemperature be estimated from ice cores?, Global Biogeochemical Cycles, 14, 851-861, 2000.

Hoffmann, G., Jouzel, J., and Johnsen, S.: Deuterium excess record from central Greenland over the last millennium: Hints of a North Atlantic signal during the Little Ice Age, Journal of Geophysical Research: Atmospheres, 106, 14265-14274, 2001.

Jouzel, J., Hoffmann, G., Koster, R., and Masson, V.: Water isotopes in precipitation:: data/model comparison for present-day and past climates, Quat. Sci. Rev., 19, 363-379, 2000.

Landais, A., Steen-Larsen, H. C., Guillevic, M., Masson-Delmotte, V., Vinther, B., and Winkler, R.: Triple isotopic composition of oxygen in surface snow and water vapor at NEEM (Greenland), Geochimica et Cosmochimica Acta, 77, 304-316, 2012.

Landais, A., Casado, M., Prié, F., Magand, O., Arnaud, L., Ekaykin, A., Petit, J.-R., Picard, G., Fily, M., and Minster, B.: Surface studies of water isotopes in Antarctica for quantitative interpretation of deep ice core data, Comptes Rendus Geoscience, 2017.

Masson-Delmotte, V., Hou, S., Ekaykin, A., Jouzel, J., Aristarain, A., Bernardo, R. T., Bromwich, D., Cattani, O., Delmotte, M., Falourd, S., Frezzotti, M., Gallée, H., Genoni, L., Isaksson, E., Landais, A., Helsen, M. M., Hoffmann, G., Lopez, J., Morgan, V., Motoyama, H., Noone, D., Oerter, H., Petit, J. R., Royer, A., Uemura, R., Schmidt, G. A., Schlosser, E., Simões, J. C., Steig, E. J., Stenni, B., Stievenard, M., van den Broeke, M. R., van de Wal, R. S. W., van de Berg, W. J., Vimeux, F., and White, J. W. C.: A Review of Antarctic Surface Snow Isotopic Composition: Observations, Atmospheric Circulation, and Isotopic Modeling*, J. Clim., 21, 3359-3387, 10.1175/2007JCLI2139.1, 2008.

Masson-Delmotte, V., Steen-Larsen, H., Ortega, P., Swingedouw, D., Popp, T., Vinther, B., Oerter, H., Sveinbjörnsdottir, A., Gudlaugsdottir, H., and Box, J.: Recent changes in north-west Greenland climate documented by NEEM shallow ice core data and simulations, and implications for past-temperature reconstructions, 2015.

NEEM: Eemian interglacial reconstructed from a Greenland folded ice core, Nature, 493, 489-494,

http://www.nature.com/nature/journal/v493/n7433/abs/nature11789.html#supplementary-information, 2013.

NorthGRIP: High-resolution record of Northern Hemisphere climate extending into the last interglacial period, Nature, 431, 147-151, http://www.nature.com/nature/journal/v431/n7005/suppinfo/nature02805_S1.html, 2004.

Oyabu, I., Matoba, S., Yamasaki, T., Kadota, M., and Iizuka, Y.: Seasonal variations in the major chemical species of snow at the South East Dome in Greenland, Polar Science, 10, 36-42, https://doi.org/10.1016/j.polar.2016.01.003, 2016.

Persson, A., Langen, P. L., Ditlevsen, P., and Vinther, B. M.: The influence of precipitation weighting on interannual variability of stable water isotopes in Greenland, Journal of Geophysical Research: Atmospheres, 116, 2011.

Pfahl, S., and Sodemann, H.: What controls deuterium excess in global precipitation?, Climate of the Past, 10, 771-781, 2014.

Risi, C., Landais, A., Winkler, R., and Vimeux, F.: Can we determine what controls the spatio-temporal distribution of d-excess and 17O-excess in precipitation using the LMDZ general circulation model?, Clim. Past, 9, 2173-2193, 10.5194/cp-9-2173-2013, 2013.

Ritter, F., Steen-Larsen, H. C., Werner, M., Masson-Delmotte, V., Orsi, A., Behrens, M., Birnbaum, G., Freitag, J., Risi, C., and Kipfstuhl, S.: Isotopic exchange on the diurnal scale between near-surface snow and lower atmospheric water vapor at Kohnen station, East Antarctica, The Cryosphere Discuss., 2016, 1-35, 10.5194/tc-2016-4, 2016.

Schotterer, U., Oldfield, F., and Fröhlich, K.: GNIP. Global Network for Isotopes in Precipitation, 1996.

Sodemann, H., Masson-Delmotte, V., Schwierz, C., Vinther, B. M., and Wernli, H.: Interannual variability of Greenland winter precipitation sources: 2. Effects of North Atlantic Oscillation variability on stable isotopes in precipitation, Journal of Geophysical Research: Atmospheres, 113, 2008.

Sokratov, S. A., and Golubev, V. N.: Snow isotopic content change by sublimation, Journal of Glaciology, 55, 823-828, 10.3189/002214309790152456, 2009.

Steen-Larsen, H. C., Masson-Delmotte, V., Sjolte, J., Johnsen, S. J., Vinther, B. M., Bréon, F. M., Clausen, H. B., Dahl-Jensen, D., Falourd, S., Fettweis, X., Gallée, H., Jouzel, J., Kageyama, M., Lerche, H., Minster, B., Picard, G., Punge, H. J., Risi, C., Salas, D., Schwander, J., Steffen, K., Sveinbjörnsdóttir, A. E., Svensson, A., and White, J.: Understanding the climatic signal in the water stable isotope records from the NEEM shallow firn/ice cores in northwest Greenland, Journal of Geophysical Research: Atmospheres, 116, n/a-n/a, 10.1029/2010JD014311, 2011.

Steen-Larsen, H. C., Johnsen, S. J., Masson-Delmotte, V., Stenni, B., Risi, C., Sodemann, H., Balslev-Clausen, D., Blunier, T., Dahl-Jensen, D., Ellehøj, M. D., Falourd, S., Grindsted, A., Gkinis, V., Jouzel, J., Popp, T., Sheldon, S., Simonsen, S. B., Sjolte, J., Steffensen, J. P., Sperlich, P., Sveinbjörnsdóttir, A. E., Vinther, B. M., and White, J. W. C.: Continuous monitoring of summer surface water vapor isotopic composition above the Greenland Ice Sheet, Atmos. Chem. Phys., 13, 4815-4828, 10.5194/acp-13-4815-2013, 2013.

Steen-Larsen, H. C., Masson-Delmotte, V., Hirabayashi, M., Winkler, R., Satow, K., Prié, F., Bayou, N., Brun, E., Cuffey, K. M., Dahl-Jensen, D., Dumont, M., Guillevic, M., Kipfstuhl, S., Landais, A., Popp, T., Risi, C., Steffen, K., Stenni, B., and Sveinbjörnsdottír, A. E.: What controls the isotopic composition of Greenland surface snow?, Clim. Past, 10, 377-392, 10.5194/cp-10-377-2014, 2014.

Steen-Larsen, H. C., Masson-Delmotte, V., Sjolte, J., Johnsen, S. J., Vinther, B. M., Bréon, F. M., Clausen, H., Dahl-Jensen, D., Falourd, S., and Fettweis, X.: Understanding the climatic signal in the water stable isotope records from the NEEM shallow firn/ice cores in northwest Greenland, Journal of Geophysical Research: Atmospheres, 116, 2011.

Stenni, B., Scarchilli, C., Masson-Delmotte, V., Schlosser, E., Ciardini, V., Dreossi, G., Grigioni, P., Bonazza, M., Cagnati, A., Karlicek, D., Risi, C., Udisti, R., and Valt, M.: Three-year monitoring of stable isotopes of precipitation at Concordia Station, East Antarctica, The Cryosphere, 10, 2415-2428, 10.5194/tc-10-2415-2016, 2016.

Touzeau, A., Landais, A., Stenni, B., Uemura, R., Fukui, K., Fujita, S., Guilbaud, S., Ekaykin, A., Casado, M., Barkan, E., Luz, B., Magand, O., Teste, G., Le Meur, E., Baroni, M., Savarino, J., Bourgeois, I., and Risi, C.: Acquisition of isotopic composition for surface snow in East Antarctica and the links to climatic parameters, The Cryosphere, 10, 837-852, 10.5194/tc-10-837-2016, 2016.

Werner, M., Heimann, M., and Hoffmann, G.: Isotopic composition and origin of polar precipitation in present and glacial climate simulations, Tellus B: Chemical and Physical Meteorology, 53, 53-71, 10.3402/tellusb.v53i1.16539, 2001.

Werner, M., Jouzel, J., Masson-Delmotte, V., and Lohmann, G.: Reconciling glacial Antarctic water stable isotopes with ice sheet topography and the isotopic paleothermometer, Nature communications, 9, 3537, 2018.

---

## Author Comment (AC1) · 5 Oct 2019

Dear Editor and Reviewers,

With this letter, we are submitting a response to reviewer comments for the manuscript (CP-2019-74) entitled "Precipitation and ice core $\delta$D-$\delta$18O line slopes and their climatological significance" for consideration for publication in Climate of the Past. We thank the reviewers for providing useful comments that will lead to an overall improvement of the manuscript. We hope that our responses have effectively addressed the criticisms of the reviewers.

Prior to addressing the individual comments, we would like to raise one broader concern to the editor. There are a number of questions raised by each reviewer related

to issues that have been previously addressed by an earlier study from some of the authors of this manuscript. The manuscript Kopec et al. (2019), of which three of the authors to this manuscript are a part, examines the precipitation isotope record at Summit that provides the basis for much of the new work presented in this manuscript. While we recognize that some of the discussion presented in this manuscript should be put more into context of the range of work that has been previously published, and we will certainly do so in a revised version, we do not think that it is necessary to rearticulate the same arguments here that are presented in Kopec et al. (2019). In our responses below we identify a number of places where a given argument by the reviewer has been previously addressed. We are certainly open to adding more discussion to this manuscript as needed to ensure we fairly acknowledge the many important studies conducted before us, but we would like to know if this approach is acceptable to the Editor.

In the Supplement Document, we provide a point-by-point response to each comment by the reviewers. We quote and italicize each Reviewer comment, and then state our responses below each comment.

Please also note the supplement to this comment:
https://www.clim-past-discuss.net/cp-2019-74/cp-2019-74-AC1-supplement.pdf

**Supplement:**

**RC1 Comments and Responses**

"*The manuscript submitted by Kopec and others deals with the relationship between d18O and dD in Greenland with new data from the Owen ice core drilled at Summit and covering the period 1977 to 2010. The main message covey by this manuscript is a discussion of the slope between dD and d18O with difference in summer and winter that is attributed to an important contribution of surface sublimation in Greenland to the precipitation at Summit in summer. The authors also propose a way to link the slope to a budget of sublimation vs precipitation amount.*

*I can not support the publication of such manuscript for many reasons given below:*
*-The authors can not ignore all the recent literature on the d18O – d-excess in surface snow and shallow firn cores showing different results that those presented here. As an example, Steen-Larsen et al. (2011) did a very detailed analysis of d18O – d-excess variations on a shallow ice core at NEEM. This study was followed by the manuscript of Masson-Delmotte, Steen-Larsen et al. (2015) with much more data. In these two manuscripts, the link between d18O and d-excess is clearly different from what is presented in the present paper. The interpretation is thus different as well, with an important contribution of marine source evaporation in the whole d-excess signal. I don't challenge the measurements performed in the present study since the water isotopic measurements are routine work but it is not correct to present new data contradicting previous recent ones in ignoring this work. This is particularly problematic since the authors propose an interpretation by using the global sublimation flux over Greenland and not any regional estimate (or calculated using backtrajectories for example) so that there is no reason why the explanation proposed in the present study should not be valid for another Greenland site.*"

- We apologize for not citing some of the work related to our study and will do so in subsequent versions of this manuscript. However, we discuss the work presented here in context of Kopec et al (2019), which does describe in great detail how the precipitation d-excess measurements at Summit differ from those at NEEM and other sites around the Arctic, and why we invoke a new mechanism - sublimation. In a revised version of the manuscript, we will ensure the language makes this point clear, and we will include discussion and citations of these studies as needed.
- The proposed explanation is certainly valid elsewhere on the ice sheet. Our SPI calculation takes into account the water vapor flux off the ice sheet and the total precipitable water at a given site. In most other locations, the precipitable water is a larger amount (since most locations are meteorologically closer to marine moisture sources than Summit), and thus this effect is minimized.
    - While sublimation contribution might be minimal elsewhere, this phenomenon does not have to be limited only to Summit. As Steen-Larsen et al. (2011) state, on average, the diffusion corrected d-excess appears to lag $\delta^{18}O$ by 4-5 months at NEEM, and thus $\delta D$-$\delta^{18}O$ slopes are less than 8. However, it can be seen in their Figure 7 that, at times, the d-excess is close to being in-phase with $\delta^{18}O$, and thus annual $\delta D$-$\delta^{18}O$ slopes are likely greater than 8 for those years. It is possible that sublimation contribution to precipitation at NEEM is significant under certain conditions. While detailed analysis of NEEM data is beyond the scope of this work, we point out that sublimation is only one potential source of moisture for all Greenland sites. Whether this signature can be singled out at each site depends on the relative importance of all potential sources.

Even in our work at Summit, the SPI and PDI only explain 61% of the total variance in slope variations. We only argue here that the signature of sublimation is sufficiently prominent to be identified.

- We would like to make an additional point regarding SPI. As presently discussed in the manuscript, SPI is described as a quantitative estimate of how much sublimation sourced moisture is contributing to Summit precipitation. We recognize that we should present the concept of SPI as more of a proxy rather than a rigorous quantitative measure of the sublimation contribution, and plan to alter the discussion in the revised version of the manuscript.

*"In the same line, Steen-Larsen, Bonne and others (see some references at the end of the review) have largely studied the imprint of evaporation over the ocean and sublimation over the snow in Greenland on the d18O, dD and hence d-excess signals both with monitoring of the water vapor isotopic composition and with modelling approaches including water isotopes. Again, the authors does not quote any of these studies and only quote for sublimation some older papers that are even not listed in the reference list (Moser and Stichler, 1974; Stichler et al., 2001)."*

- The study by Steen-Larsen et al. (2014), which is discussed in Kopec et al. (2019), presents evidence that sublimation caused isotopic change to the remaining snow, but the changes were not systematic (due to a variety of other factors, including the surface water vapor isotopic composition, rate of wind pumping, and the temperature gradient in the snow) and thus did not reach any conclusion on how snow fractionates during sublimation. In fact, they discuss the need for "controlled laboratory experiments and isotopic modeling" to better constrain these mechanisms. In a constrained laboratory study like the one presented by Moser and Stichler (1974), which we cited, they can more readily isolate isotopic changes due to fractionation during mass loss by sublimation, in which they show a reduction of d-excess of the snow (and thus an addition of relatively high d-excess water vapor to the atmosphere). We also cited a study containing isotopic modeling of sublimation, the study by Stichler et al. (2001), which was able to effectively calculate the changes observed in the snowpack.
- Two of the studies that were mentioned by the Reviewer just came out a few months ago – Bonne et al. (2019) and Madsen et al. (2019), which we did not address in the previous version of the manuscript but will do so in the revised version. Both studies show that significant isotopic change occurs during sublimation. Bonne et al. show that the sublimation of snow over sea ice produces water vapor with relatively high d-excess, especially compared to that which would be sourced from the cold ocean surface below. A straightforward assumption can be made that sublimation of snow on the ice sheet would also produce high d-excess. Madsen et al. show that significant sublimation and deposition take place over eight diurnal cycles, which alters the isotopic composition significantly. Although not stated explicitly in this manuscript, it appears that from their Figure 2, when the latent heat flux is positive (sublimation is taking place), the water vapor d-excess is higher than that when the latent heat flux is negative (deposition). This is consistent with the findings of Bonne et al. and what we inferred from Moser and Stichler. We will describe these findings in the revised manuscript.

*"In addition to the recent literature, older papers are also fully ignored such as the study of Hoffmann et al. (2001) presenting a fully different interpretation of the recent d-excess signal at GRIP, i.e. at Summit, and a different signal too."*

-   The study by Hoffmann et al. (2001) will be cited and discussed in the revised version of the manuscript. However, this study is not a relevant comparison to our analysis for two main reasons. 1) It only discusses longer term variations and not annual/seasonal variations, the timescales we focus on. 2) Their discussion focuses on marine source variations in the North Atlantic that cause the changes of d-excess observed in the core and bases the analysis on the classic Merlivat and Jouzel evaporation models. As discussed at length by Kopec et al. (2019), the type of explanation in Hoffman et al. cannot account for the precipitation d-excess annual cycle at Summit in their study.

*"The dating of the Owen ice core is not described sufficiently while the whole analysis is dependent on this dating. A whole section should be devoted to this aspect showing the chemical concentrations, how they are used to date the ice. In the present manuscript, this section is not robust enough to support the conclusion."*

-   As we write in the manuscript, the dating is done using $\delta^{18}O$. It is reaffirmed with chemical concentration measurements, but we strictly use the isotopic measurements to delineate each year. The dating analysis we employ here is quite standard and is how it is done in manuscripts cited by the reviewer, including Steen-Larsen et al. (2011) and Masson-Delmotte et al. (2015). As can be seen in the data presented in our Figure 2, the annual cycle of $\delta^{18}O$ is extremely clear (with the exception of two years, in which we gave a more detailed description in the original version of the manuscript), and thus the dating is quite straightforward. In the revised version of the manuscript, we will expand upon this section to more fully describe how we delineate different years to ensure greater clarity on this process.

*"I am very concerned about the way diffusion is treated in this paper. Even if we consider the simple diffusion model of Johnsen correct, it is not used in the right way here. Indeed, in the initial paper by Johnsen et al. (2000), it is stated on section 2.2.3 (p. 171) that an artificial signal of d-excess is created by diffusion and this is observed in the figure 4 of this paper. In other words, the diffusion does not only play a role in the amplitude of the d-excess signal as mentioned and calculated in this study but also on the phasing between the d18O and d-excess signals. Steen-Larsen et al. (2013) used the Johnsen model and corrected then for a phasing between d-excess and d18O on the NEEM shallow ice core. I am very surprised that the authors do fully ignore this effect which probably fully biases their analysis and interpretation. I am also very surprised that the authors only show the raw series of d18O and d-excess and never the diffusion corrected series.*"

-   We disagree with the premise of this comment; we correctly account for the phase shift of d-excess in our analysis. The phase change of d-excess is a result of the change in the amplitude ratio of $\delta D$ over $\delta^{18}O$. The d-excess value is calculated from $\delta D$ and $\delta^{18}O$, and thus the effect of diffusion on d-excess is dealt with by calculating the diffusion effect on the amplitude of the $\delta D$ and $\delta^{18}O$ annual cycles, and thus is presented as the change of slope. In the original version of

the manuscript, we account for this change. Our Eqn 3 skips the step of showing the calculated diffusion effect for each variable, and instead combines them to directly calculate the slope change. Steen-Larsen et al. (2011) also use this same correction model determined by Johnsen et al. (2000) and thus ultimately correct the d-excess phase in a similar manner.

- o The changes to the d-excess phase by diffusion, and thus to the slope, can be seen in the results of the original manuscript. If diffusion over time caused d-excess to be in phase with $\delta^{18}O$, the slope of the $\delta D$-$\delta^{18}O$ line would be artificially increased, and thus removing the diffusion effect should reduce the slope. The time series of annual $\delta D$-$\delta^{18}O$ slopes in Figure 3 show that the diffusion correction reduces the slope the greatest for the oldest measurements, reducing raw slope values over 9 to slopes below 8. This slope change is equivalent to shifting d-excess from being in-phase with $\delta^{18}O$ to 180 degrees out-of-phase with $\delta^{18}O$.

- While the reviewer does not point this out, we realized that we did not state an implicit assumption in our discussion of the diffusion correction in the manuscript, which we will add to the revised version. The assumption we make in the calculation of the diffusion effect on the slope (our equation 2) is that $\delta D$ and $\delta^{18}O$ are in phase. In other words, when $\delta D$ and $\delta^{18}O$ are in phase, and this in-phase relationship does not change with diffusion, then the slope of $\delta D$ vs. $\delta^{18}O$ (d$\delta D$/d$\delta^{18}O$) for a sinusoidal cycle is the amplitude ratio of $\delta D$ over $\delta^{18}O$. We had observationally confirmed that this was true for both precipitation data and the ice core data.

However, this reviewer's comment challenged us to consider this assumption with more rigor. We have analyzed the annual phase relationship between $\delta D$ and $\delta^{18}O$ for both precipitation data and the ice core data. For precipitation, the phase difference between $\delta D$ and $\delta^{18}O$ ranges from -2.20 to +6.05 days, with the mean of 2.94 and standard deviation 3.57 days. The phase difference for any given year or for the average is not significantly different from zero (p = 0.20 for the average). For the ice core the phase difference ranges from -10.1 to +8.2 days, with the mean of 1.15 and standard deviation of 4.76 days. The phase difference of individual years and the average is again not significantly different from zero (p = 0.18 for the average). While it is difficult to theoretically establish that the phase difference between $\delta D$ and $\delta^{18}O$ does not change by diffusion, we consider it adequate to assume that the phase difference is sufficiently close to zero both before and after diffusion, given the above analysis and many published observations for precipitation and ice cores where $\delta D$ and $\delta^{18}O$ linearly covary.

To be further cautious, we assessed the error of the calculated slope when the phase difference is not zero. For the actual best fit phase differences, the error introduced to the slope calculations ranges from 0.00 (<1 day) to 0.38% (10 days). In addition, we conducted a correction of the annual slope estimates for the ice core using the best fit annual phase differences between $\delta D$ and $\delta^{18}O$, and reconducted the analyses in Figures 3 and 5. The results remain the same. Therefore, we consider our results robust. Since none of the phase differences are significantly different from zero, we feel that it is better not to do any correction (keep the analysis as is in the original manuscript) because statistically it does not yield significantly different results. However, we would be pleased to incorporate this information in a Supplementary material, if so requested.

*"Summarizing, I have serious doubts on the robustness of the dating and diffusion correction of the series presented here to follow the interpretation proposed. Moreover, the ignorance of a rich and documented literature on the subject limits the scientific interest of the present study for a large community working on water isotopes in the high latitudes of the northern hemisphere."*

- We hope that our explanation above adequately addresses the issues raised by the reviewer. While we believe many of the criticisms raised by the reviewer based on the suggested lack of acknowledgment of previous work are dealt with in the discussion of the precipitation d-excess data in Kopec et al. (2019), which provides the basis of much of the new work presented here, we do acknowledge that these earlier studies should be included in this manuscript to provide a clearer context for this work.

*"References cited by Reviewer 1:*

*Bonne, J. L., Behrens, M., Meyer, H., Kipfstuhl, S., Rabe, B., Schönicke, L., ... & Werner, M. (2019). Resolving the controls of water vapour isotopes in the Atlantic sector. Nature communications, 10(1), 1632.*

*Hoffmann, G., Jouzel, J., & Johnsen, S. (2001). Deuterium excess record from central Greenland over the last millennium: Hints of a North Atlantic signal during the Little Ice Age. Journal of Geophysical Research: Atmospheres, 106(D13), 14265-14274*

*Johnsen, S. J., Clausen, H. B., Cuffey, K. M., Hoffmann, G., Schwander, J., & Creyts, T. (2000). Diffusion of stable isotopes in polar firn and ice: the isotope effect in firn diffusion. In Physics of ice core records (pp. 121-140). Hokkaido University Press.*

*Madsen, M. V., Steen-Larsen, H. C., Hörhold, M., Box, J., Berben, S. M. P., Capron, E., ... & Kipfstuhl, S. (2019). Evidence of isotopic fractionation during vapor exchange between the atmosphere and the snow surface in Greenland. Journal of Geophysical Research: Atmospheres, 124(6), 2932-2945.*

*Masson-Delmotte, V., Steen-Larsen, H. C., Ortega, P., Swingedouw, D., Popp, T., Vinther, B. M., ... & Falourd, S. (2015). Recent changes in north-west Greenland climate documented by NEEM shallow ice core data and simulations, and implications for past-temperature reconstructions.*

*Steen-Larsen, H. C., Masson-Delmotte, V., Sjolte, J., Johnsen, S. J., Vinther, B. M., Bréon, F. M., ... & Gallée, H. (2011). Understanding the climatic signal in the water stable isotope records from the NEEM shallow firn/ice cores in northwest Greenland. Journal of Geophysical Research: Atmospheres, 116(D6)*

*Steen-Larsen, H. C., Masson-Delmotte, V., Hirabayashi, M., Winkler, R., Satow, K., Prié, F., ... & Dumont, M. (2014). What controls the isotopic composition of Greenland surface snow?. Climate of the Past, 10(1), 377-392*

*Steen-Larsen, H. C., Risi, C., Werner, M., Yoshimura, K., & Masson-Delmotte, V. (2017). Evaluating the skills of isotope-enabled general circulation models against in situ atmospheric water vapor isotope observations. Journal of Geophysical Research: Atmospheres, 122(1), 246-263."*

---

## Author Comment (AC2) · 5 Oct 2019

Dear Editor and Reviewers,

With this letter, we are submitting a response to reviewer comments for the manuscript (CP-2019-74) entitled "Precipitation and ice core $\delta$D-$\delta$18O line slopes and their climatological significance" for consideration for publication in Climate of the Past. We thank the reviewers for providing useful comments that will lead to an overall improvement of the manuscript. We hope that our responses have effectively addressed the criticisms of the reviewers.

Prior to addressing the individual comments, we would like to raise one broader concern to the editor. There are a number of questions raised by each reviewer related

to issues that have been previously addressed by an earlier study from some of the authors of this manuscript. The manuscript Kopec et al. (2019), of which three of the authors to this manuscript are a part, examines the precipitation isotope record at Summit that provides the basis for much of the new work presented in this manuscript. While we recognize that some of the discussion presented in this manuscript should be put more into context of the range of work that has been previously published, and we will certainly do so in a revised version, we do not think that it is necessary to rearticulate the same arguments here that are presented in Kopec et al. (2019). In our responses below we identify a number of places where a given argument by the reviewer has been previously addressed. We are certainly open to adding more discussion to this manuscript as needed to ensure we fairly acknowledge the many important studies conducted before us, but we would like to know if this approach is acceptable to the Editor.

In the Supplement Document, we provide a point-by-point response to each comment by the reviewers. We quote and italicize each Reviewer comment, and then state our responses below each comment.

Please also note the supplement to this comment:
https://www.clim-past-discuss.net/cp-2019-74/cp-2019-74-AC2-supplement.pdf

―――――――――――――――――――――

[Figure]

**Supplement:**

**RC2 Comments and Responses**

*"This paper presents new measurements of water isotopic composition from the site of Summit in Greenland where several ice cores have been drilled. The results compare precipitation isotopic composition to an ice core record and highlight a peculiar relationship between $\delta^{18}O$ and $\delta D$ (greater than 8), leading to values of d-excess singularly lower to what is usually found in Greenland. They interpret this different behaviour to the impact of sublimation on the snow isotopic composition, and thus suggest that a large part of the d-excess signal represent the sublimation over the ice sheet, and not the evaporation conditions as commonly interpreted.*

*I read the manuscript with great interests, it rises an interesting and plausible alternative explanation of the slope between $\delta^{18}O$ and $\delta D$ affecting the interpretation of the isotopic paleothermometer. However, I have several concerns of major and minor nature, which address a few methodological aspects as well as the description of the relevant processes.*

*General comments:*
*1. In this manuscript, the authors suggest that the slope between $\delta^{18}O$ and $\delta D$ found at summit is an anomaly basing their studies on the results of Feng et al, 2009. The new results from summit seems to present positively correlated variations of d-excess against $\delta^{18}O$ while GNIP data (Feng et al, 2009) indicate that the normal behaviour is negatively correlated at the seasonal scale.*

*The negatively correlated relation between d-excess and $\delta^{18}O$ is normal for low isotopic composition areas due to the Rayleigh distillation. Typically, for a Rayleigh distillation process, the local variations of and can be calculated by:*

$$\frac{d\delta D}{d\delta^{18}O} = \frac{\alpha_D - 1}{\alpha_{18} - 1}\frac{1 + \delta D}{1 + \delta^{18}O}$$

*where $\alpha_{18}$ and $\alpha_D$ are the respective equilibrium fractionation coefficients (or effective if you include kinetic fractionation). For very low temperature conditions, $\alpha_D\text{-}1/\alpha_{18}\text{-}1 \approx 10$ (typically for -30°C, you would have 9.65). Yet, for low isotopic composition such as found at summit ($\delta D \approx$ -300‰, $\delta^{18}O \approx$-35‰), This means the expected slope between $\delta^{18}O$ and $\delta D$ would be 7‰/‰. This effect has been described in multiple papers such as (Touzeau et al., 2016;Casado et al., 2016).*

*This generalise to any time scales the results of Feng et al (2009), and thus is key for your manuscript where you compare precipitation isotopic composition at the seasonal scale to an ice core records spanning more than 30 years. It also shows that in both winter and summer, the slopes you obtained are larger than expected by the Rayleigh distillation."*

- It is not clear to us what this reviewer is trying to say. Do they mean that we obtained seasonal slopes that do not agree with the expected slope of Rayleigh distillation and we should address that, or do they mean that we, or Feng et al. (2009), are over-interpreting the observed d-excess vs. $\delta^{18}O$ relationship? In either case, we argue that the simple relationship for the slope given by the reviewer is just an approximation. First, its derivation is based on constant-alpha ($\alpha$, the fractionation factor) Rayleigh distillation, which never happens in nature. Second, the derivation

from the Rayleigh distillation curve to this expression again assumes that alpha values are constant with respect to F (the fraction of distillation). Since d-excess is a very small quantity compared to $\delta D$ and $\delta^{18}O$, these assumptions result in significant errors in the slope estimate.

- Additionally, the effect of Rayleigh distillation on precipitation at Summit, Greenland is discussed at length in Kopec et al. (2019). As described in Kopec et al. (2019) and in what is presented in this manuscript, Rayleigh distillation can explain some of, but not all of, the relationships observed in the precipitation isotopic data.

*"2. There are previous observations of positively correlated d-excess and $\delta^{18}O$ (or $\delta D$) in Greenland. For instance, Barlow et al. (1993) present data from Summit where $\delta D$ and d-excess are correlated at the seasonal scale in an ice core. This seesaw variation was later attributed before to variations of SST in the north atlantic (Hoffmann et al., 2001) using a Rayleigh distillation model. In depth discussion on the difference between your results and their results are necessary."*

- While we did not cite either of these manuscripts here, the concepts presented in those manuscripts and written here by the reviewer were discussed at length in Kopec et al. (2019), where marine source variations and Rayleigh distillation cannot explain the variations we observe at Summit, particularly the anomalously high summer d-excess. In the revised version of the manuscript, we will cite these studies and more clearly emphasize our point.

*"3. Section 4.1 is essentially a second introduction. I recommend that you to consider integrating it into the introduction and removing it from here.*

*Oppositely, what is missing from the discussion is the comparison of your results with time series and slopes from other study at summit and in other neighbouring sites in Greenland."*

- We choose to include this information in 4.1 rather than in 1. Introduction, and prefer to keep it in this section, because the discussion of slope here is written in the context of our result that the $\delta D$-$\delta^{18}O$ slope is greater than 8, rather than the typical one less than 8 that is observed at most sites around the world. Why slopes are less than 8 is discussed in the introduction in the context of seasonally changing d-excess in association with changing marine moisture sources. In other words, the focus there is seasonal variation of d-excess. But here the focus is the slope itself; we attempt to describe all possible factors that can affect the slope. In the revised manuscript, we can certainly make more clear connections between the d-excess annual cycle and the $\delta D$-$\delta^{18}O$ slope in the Introduction, while keeping the detailed discussion of slope variations where it is now. Alternatively, we can move all the material in 4.1 to 1, but this may make the introduction too long, and less focused. We prefer the current organization, but are willing to change if this reviewer and editor feel strongly that we should do so.
- Regarding discussing slopes in context of neighboring sites, we will include some discussion in a revised manuscript to address how Summit $\delta D$-$\delta^{18}O$ slopes are similar or different than nearby locations, such as at NEEM as observed in Steen-Larsen et al. (2011). We emphasize that the $\delta D$-$\delta^{18}O$ slopes of neighboring sites support the main argument and conclusion of our paper.

*"4. Section 4.2 is essentially a method sub-section, except for the first two paragraphs which are introduction material. Consider integrate it into the methods and removing it from here."*

- We disagree with this point. We consider the Monte Carlo simulations a test of our hypotheses that are not proposed until discussion. Most of the remaining components of the paragraph provide the setup for that test, and are not introduction material. We are sympathetic that this reviewer may be frustrated by the detailed description of how we did it, but moving this material to the Method and/or Introduction section would lose its context. We prefer to keep it this way, but are willing to consider moving it to an appendix if so demanded.

*"5. The arguments in section 4.3, which are central to the arguments of this manuscript, are developed with unsatisfactory level of details.*

*First, it is not clear what data are used to calibrate the multi-linear regression problem in section 4.3.3. and in Fig. 5. Only 16 diffusion corrected slope data points appear on the plots while you initially have 32 years obtained from the owen ice core. Were the data selected to cover the period observed in Box et al, 2006 ? If yes, is this period representative of the entire period ?"*

- Yes, this is the period observed in Box et al. (2006). This will be made more explicit in the revised version of the manuscript. The study by Box et al. covers approximately the middle 17 years of our 32-year analysis. The observed $\delta D$-$\delta^{18}O$ slopes during this time window are similar to the average of the whole dataset. Thus, we consider this period representative and will emphasize this point in the revised manuscript.

*"Second, is the SPI over the entire surface of Greenland representative of the SPI across the area where moisture travels from the North Atlantic to Summit? As there is a significant surface north of Summit where one does not expect any summer precipitation travel path to go to, it would be important to evaluate how this impacts the data. Are the precipitation also coming more generally from trajectories from the Davis Straight of the Denmark Straight? Considering that the patterns observed are very different for the South East, the South West, and the North, one would expect this to impact the results. In general, changes of origin of the air masses from the west or the east of Greenland would be expected to impact the results, and is not discussed here relatively to the impact of sublimation."*

- We agree that there are uncertainties in the representativeness of SPI to capture the moisture sublimating from the ice sheet and traveling to Summit. As presently discussed in the manuscript, SPI is described as a quantitative estimate of how much sublimation sourced moisture is contributing to Summit precipitation. We recognize that we should present the concept of SPI as much more of a proxy of this contribution rather than a rigorous quantitative measure of the sublimation contribution, and plan to alter the discussion in the revised version of the manuscript.
- Regarding the transport of moisture from marine sources, it is discussed at length in Kopec et al. (2019) how marine sourced moisture alone cannot explain the observed annual cycle of d-excess at Summit. This index is not refined by the moisture source and path, but reflects, on average, interannual variations of the potential for sublimation moisture to contribute to

Summit precipitation. The question by the reviewer asking which of these pathways the moisture takes from the marine source, or additionally which of these pathways potentially incorporate more sublimated moisture, is certainly an interesting one, and would be important in ultimately quantifying the sublimation contribution. However, these differences are beyond the scope of our discussion here, and thus not included.

*"Third, how good is ERA-interim precipitable water amount? In general, the amount of precipitation over the ice sheets is quite biased."*

- Quantifying the ability for ERA-Interim to accurately represent the precipitable water amount is not in the scope of this study, but it is one of the most commonly used reanalysis products, and so we chose to use this dataset. Another commonly used reanalysis dataset is the NCEP/NCAR Reanalysis product. The precipitable water estimate from ERA-Interim is quite similar to that estimated by the NCEP/NCAR Reanalysis product. Additionally, ERA-Interim data gives us a higher spatial resolution than the NCEP/NCAR Reanlaysis data, so we decided to use ERA-Interim to give us the best estimate of the precipitable water at Summit. However, we recognize that there is uncertainty in these reanalysis products and we will add discussion of this point in the revised manuscript

*"There is no interpretation of the various regressions in Fig. 5. For instance, earlier in the manuscript, you describe the temporal slope of the diffusion corrected slopes as not significant. So how are the trends compared to the original trend?"*

- We wonder if the reviewer may have confused Figure 3 with Figure 5. Figure 3 is the temporal trend, while Figure 5 is the result of the multiple regression. Each panel of Figure 5 is part of a multiple regression showing the effect of each of the three explanatory variable leverages labelled as the x-axes, which is described in section 4.3. These are not plots of slope over time; those plots are in Figure 3.

*"It seems also that the vertical axes of the three different sub-figures of figure 5 are different as the horizontal levels of the larger and smaller vertical values do not match (I can't judge for the other points)."*

- The scale of the vertical axes is the same in each leverage plot. The vertical values shown in each plot differ slightly as the leverage plots show the relationship between a given explanatory variable and the response variable with the effect of the other explanatory variables removed. See Sall (1990) for a discussion on how these leverage values shown in Figure 5 are calculated.

*"How crucial is the diffusion correction for these results?"*

- Without the diffusion correction, the results are still significant (overall $r^2$ = 0.67, p = 0.0018) and each partial coefficient is significant (p < 0.05) and in the same direction. This suggests that the results are robust even if there is uncertainty in the diffusion correction. We will provide addition discussion of this point in the revised version of the manuscript.

*"How good would be a prediction from the multi-linear regression using the best fit to actually predict your slope time series?"*

- Figure 5 and Figure 3 are two different regressions. One cannot easily estimate the error of one regression using the statistics of another. The multiple regression in Figure 5 yields an $r^2 = 0.61$, indicating that the regression explains 61% of the total variance in the slope variations. This 61% variation includes both interannual variations (which does not contribute to the trend) as well as the temporal variations (that does include the trend), but primarily the former. In addition, after diffusion correction, the temporal trend is not statistically significant, and thus we did not pursue it further.

*"6. It is not clear to me how you discard the change of trajectory of the air masses between summer and winter, which could also explain the offset between summer and winter values.*
*For instance, have you performed any back-trajectory analysis that shows that for the period from 1979 to today, the air masses originates and travel roughly across the same areas and are under the same level of distillation?"*

- In Kopec et al (2019), the authors discuss at length how changing marine-sourced air mass trajectories are unlikely to create the offset between summer and winter values. They also perform backtrajectory analysis of the precipitation data from 2011 to 2014 that shows trajectories are consistent with sublimation contribution. Since we do not have storm by storm data all the way over the time period of the ice core, we assume that the same processes are at work. We will make this assumption more explicit in the revised version of the manuscript.

*"7. As you mention the large impact of sublimation across the Greenland ice sheet to the moisture that leads to the formation of precipitation, I think you need to mention the impact on the surface snow (see the comments of the other reviewers)."*

- The isotopic compositions of the surface snow would definitely be impacted by sublimation. However, assessing this impact is not trivial and is beyond the scale of this study. To quantify the changes in surface snow, one would need to do high-resolution sampling of the surface snow. Because of wind-blown mixing and new snow deposition, it is not easy to tease out the sublimation signal. We hope that future investigations can be directed to this question, which will provide additional support to this study. This point is discussed in Kopec et al. (2019); and will also be expanded upon here in a revised version of the manuscript.

*"8. Alternatively, there are measurements of vapour isotopic composition at Summit (Berkelhammer et al., 2016) which would provide an estimation of the vapour d-excess value which can be of interest to validate the hypothesis that sublimation from the ice sheet provides a significant amount of moisture into the precipitation. The impact of the exchanges between the snow and the vapour could also affect the surface d-excess."*

- Berkelhammer et al. (2016) only present measurements of $\delta^{18}O$ and not $\delta D$ or d-excess so we cannot make estimates of sublimated vapor values to compare to our analysis.
- Conversely, Bailey et al. (2015) present water vapor isotope measurements from Summit that includes d-excess. These water vapor measurements show that d-excess is largely in phase with $\delta D$ and $\delta^{18}O$, and thus suggest a slope that is greater than 8. However, the standard error of d-excess measurements is large during the summer and presented at a low resolution, and thus prevent us from extracting the sublimated moisture d-excess end members. These findings are discussed in Kopec et al. (2019), but some additional discussion will be added to the revised version of this manuscript.

*"9. The study presented here focuses mainly on seasonal and interannual time scales. The previous study of the authors (Kopec et al, 2019) focuses on the synoptic time scale. In general, I do not believe it is possible to generalise the results from these time scales to large time scale without further studies and recommend caution to the authors to really assess at which time scales they results can be applied."*

- The primary results and discussion in Kopec et al. (2019) focus on seasonal/annual timescale where the authors explore the anomalously high summer d-excess values in the same storm-by-storm precipitation data that is presented here. We are discussing the exact same timescales here as was presented in that manuscript.

*"10. In general, the authors need to include a greater representation of the relevant literature. There are a lot of papers that would be relevant for this study that have been overlooked, which as a result weaken the manuscript."*

- The relevant literature will be discussed at a greater extent in a revised version of the manuscript.

*"Specific comments:*
*Page 2 – Line 17: "Over the past half century, variations of hydrogen (δD) and oxygen (δ18O) isotopic ratios of precipitation have served as increasingly powerful tools in a wide range of disciplines of research, including paleoclimate, hydrology, and atmospheric sciences."*

*While this is generally true, there is a large body of litterature, including review papers that you can cite here to your point…"*

- We are not sure what literature this reviewer had in mind in this particular context. We did cite 15 studies in this first paragraph. We will go through the reference list provided by this reviewer and add the relevant ones that we missed in the original manuscript.

*"Page 2 – Line 19: "One of the most striking features of these two paired isotope ratios is that they are remarkably well correlated over time and space, and this relationship is defined as the meteoric water line (MWL)."*

*At this stage, one of the early reference could also be used, for instance: (Dansgaard, 1964). You could also add one of the GNIP paper (Schotterer et al., 1996)."*

- We cite Dansgaard (1964) three sentences later when introducing the concept of deuterium excess, which is one of the primary outcomes of that study. We cite Craig (1961) when discussing the meteoric water line in the quoted sentence as that is one of the primary outcomes by Craig. We will add Schotterer et al. (1996).

*Page 2 – line 26: "In the field of paleoclimate studies, for example, δD or δ18O variations in ice cores have been used to infer temperature changes at ice core drilling sites (e.g. Jouzel et al., 1987; Dansgaard et al., 1989; Johnsen et al., 1995, 2001; Blunier et al., 2001; Petit et al., 1999) and deuterium-excess variations in ice cores have been used to infer changing marine moisture source conditions over a range of time scales (e.g. Johnsen et al., 1989; Barlow, et al., 1993; Vimeux et al., 1999, 2001; Uemura et al., 2004, 2012; Masson-Delmotte, et al., 2005; Jouzel et al., 2007)."*

*For temperature reconstruction, there are a large number of recent reference that could be included, including from Greenland: (NEEM, 2013;NorthGRIP, 2004)…*

*For the d-excess variations and the link with marine moisture source conditions, I would also recommend including more recent references*

- We used "e.g." to suggest that this is only a subset of the literature. While it is impossible to include all literature related to this subject, we will add a few recent ones as suggested.
-
*"Page 3 – Line 1: "There are relatively fewer studies focusing on how the slope of the MWL changes over space and time, and determining how such variations may contain climate information."*

*This sentence needs to be more precise:*
*1. the slope of the MWL in the isotopic composition of precipitation is what you're focusing on, as the MWL slope can change in the snow for instance due to diffusion, and thus is not containing any climatic information*
*2. Fewer than what ?*
*3. I'm not convinced that there are few studies which have looked at the spatial variations of the slope $δ^{18}O$ vs δD, for instance (Masson-Delmotte et al., 2008;Touzeau et al., 2016;Landais et al., 2017;Werner et al., 2018;Jouzel et al., 2000), for the temporal variations (Oyabu et al., 2016;Masson-Delmotte et al., 2015;Werner et al., 2001;Steen-Larsen et al., 2011;Persson et al., 2011) and for the space-time (Hendricks et al., 2000;Risi et al., 2013;Sodemann et al., 2008;Ekaykin et al., 2002)"*

- 1. Any climate proxy is affected by non-climate variables and thus have systematic and random noise. Does that mean we should not use proxies at all in climate studies? Slopes can be altered by diffusion, but that does not mean they contain no climate information.
- 2. This is referring to the (only) paragraph immediately preceding this sentence, in which we cited abundant uses of δD, $δ^{18}O$ and d-excess. We will make this clearer in the revised version.

- 3. It is true that there are not "few" studies on the slope of the δD-δ18O line, however we wrote "fewer" relative to individual isotope ratios discussed in the prior paragraph. However, we will cite more manuscripts in this paragraph as many of these studies are relevant to the work presented here.

*"Page 3 – Line 3: "For example, if the slope of a LMWL with seasonally resolved observations is less than 8, then d-excess and δD (or δ18O) would have an anti-phase relationship, and the opposite is also true "*

*I believe this is generally true, regardless of the seasonal resolution."*

- In the revised version, we will start with the general statement, and then give a specific example of the season cycle.

*"Page 3 – Line 6: "This out-of-phase relationship between the two is equivalent to the fact that, in these locations, the LMWL (with a monthly or higher resolution) has a slope less than 8. "*

*Here, the relationship is in "phase opposition", not out of phase. Out of phase could mean everything as completely random, to in phase opposition."*

- We will clarify that in the revised manuscript.

*"Page 3 – Line 10: "To our knowledge, this is the only work that described the slope distribution of LMWLs on a hemispheric scale and discussed the climatological significance of these slopes."*

*This is a very strong statement. For instance, this study (Pfahl and Sodemann, 2014) seems to do something quite similar using d-excess, which is equivalent to the slope δD-δ18O."*

- Yes, this is a fair statement that many of the same arguments can be seen in Pfahl and Sodemann (2014). We will change our language in this statement and acknowledge their work as well.

*"Page 3 – Line 11: "Temporal changes, e.g., seasonal or inter-annual, in the δD-δ18O relationship for a given location have not been explored and can potentially provide information and understanding of seasonal or interannual variations in the planet's climate system."*

*There is body of research exploring for a given location the δD-δ18O relationship. In Antarctica: (Touzeau et al., 2016;Dittmann et al., 2016;Stenni et al., 2016). In Greenland: (Landais et al., 2012;Steen-Larsen et al., 2011)"*

- We thank this reviewer for pointing to us many publications we should acknowledge. These studies do discuss controls of the δD-δ18O slope, especially on the seasonal scale, and should be cited in our study. However, they do not discuss observations of inter-annual slope changes. We will modify our language here to include these studies and discuss the difference between these studies and ours.

*Page 3 – Line 14: "Summit, Greenland is one of the most important sources of deep ice cores that provide valuable paleoclimate records, particularly through the measurement of water isotopes. "*

*I think the term "most important source of deep ice cores" is not very precise. Please reformulate.*

- We will reformulate this statement to be more specific in the revised manuscript.

*"Page 3 – Line 17: "While Feng et al. (2009) demonstrated that almost all sites in the mid- to high-latitudes of the Northern (and Southern) hemisphere exhibit an out-of-phase relationship between δD and d-excess, and the mechanisms controlling this pattern are relatively well understood, Kopec et al. (2019) recently reported a nearly in phase relationship between δD and d-excess of event-based precipitation measurements at Summit, Greenland."*

*The demonstration from Feng et al (2009) is at the seasonal scale from GNIP data. δD and d-excess are there anticorrelated.*

*Your previous study (Kopec et al, 2019) presents results across the synoptic scale and the seasonal scale.*

*Both results are not necessarily opposed considering the different time scales."*

- Both studies examine the seasonal scale. Although Kopec et al. (2019) studied data that were obtained for events, the correlation of δD and $δ^{18}O$ largely represents the seasonal co-variations of these two variables. The discussions in Kopec et al. are all focused on seasonal scale processes, so we disagree that there is a difference in time scale.

*"Page 6 – Line 22: "This result is consistent with the observations of Kopec et al. (2019), where they reported dexcess measurements in phase with δD or δ18O values."*

*Why don't you present the slope and the correlation? "In phase" is vague to describe the link between d-excess and δD."*

- The precipitation dataset is the same in Kopec et al. (2019) and in this manuscript. The phase is calculated and presented in Kopec et al. The slope, which is based on the correlation, is presented in this manuscript.

*"Page 7 – section 3.2: As you are comparing results obtain from precipitation samples and from an ice core, it would be important to transpose the diagnosis made in section 3.1 to section 3.2. In particular, I think presenting the equivalent of figure 1.b) for the Owen ice core would be beneficial. In the precipitation, you show that the slopes are actually smaller than 8 if you look at winter and summer separately, but there is a shift between the cloud of winter points and the cloud of summer points. Is the same shift visible in the Owen ice core if you do the slope on all the summer points (high $δ^{18}O$ points) and all the winter points?"*

- This is a good question, and something that we have considered. However, for the ice core, the within-year dating is much less precise so separating measurements into bins of summer and winter samples has substantial uncertainty as opposed to precipitation samples where we know the exact date the snow fell.

  However, we have examined the seasonal separation between samples that are estimated to represent summer and winter. If we attempt to recreate Figure 1 for the ice core data, the d-excess vs $\delta^{18}O$ relationship looks quite similar to that of precipitation (see Figure r1 below). We see that the summer data is separated from the winter data where the summer values are more enriched and/or have higher d-excess than those of the winter samples. To do this analysis, we assume constant snow accumulation within each year in the ice core and thus each sample within a year represents the same amount of time. From there we can define the year fraction and delineate summer and winter with the same year fraction we used to separate summer and winter samples for the precipitation dataset (i.e. summer = June, July, and August, or from year fraction 5/12 to 8/12). Given the uncertainty in defining the time of year, we would prefer not to include this figure in our revised version of the manuscript, but would be willing to include as Supplemental material if desired.

[Figure]

**Figure r1:** Plot of d-excess vs. $\delta^{18}O$ for Owen ice core dataset broken down seasonally into summer (red), winter (blue), and spring and fall (gray) points. The annual regression line is shown (black line) in addition to seasonal regression lines for summer (red line) and winter (blue line), respectively.

*"Page 7 – Line 23: "Correction for the diffusion effect on the isotopic record produces some significant differences, most prominently, the elimination of the temporal trend in the δD-δ18O slope"*

*Did you realise any sensitivity tests for this? Considering this is widely used after, it would make sense to make sure that this is pertinent.*

*Also, considering other ice core have been drilled at summit in the 90's, can you compare the values with these ice cores? It would be interesting as then, the values at the top of these old ice core would not have had time to be diffused, and thus, you can validate how much the back diffusion is not creating any artefacts."*

- As stated above in response to an earlier comment, the multiple regression of SPI, PDI, and SPI*PDI is robust across a range of diffusion corrections. It is certainly possible the tuning parameters for the diffusion correction defined by Johnsen et al. (2000) have changed, or will change for future analyses, but it is beyond the scope of this study to re-examine these factors, particularly as the primary analysis of this study appears to be relatively robust against variations in those parameters.

*"Page 10 – Line 5: "Sublimation from the snow surface has been shown to reduce the d- excess of the remaining snow, while the vapor removed by sublimation has a high d-excess (Moser and Stichler, 1974; Stichler et al., 2001)."*

*There are more recent studies studying these processes (Sokratov and Golubev, 2009;Steen-Larsen et al., 2013;Steen-Larsen et al., 2014;Casado et al., 2016;Ritter et al., 2016)"*

- We agree that some additional discussion on the impact to the surface snow can be helpful. We will incorporate the conclusions of these studies into the revised discussion.

*"Page 13 – Line 15: As figure 6 does not have a colour scale for the $\delta D$-$\delta 18O$ slope, it is very difficult to understand this paragraph. In general, I would recommend a more detail explanation."*

- We will add a statement describing the contour plot in the caption of Figure 6 in a revised version of the manuscript.

*"Page 13 – Line 31: "First and foremost, we show that the slopes of $\delta D$-$\delta 18O$ lines observed over different timescales and from various records (i.e. precipitation or ice cores) can be valuable tools to explore hydrological processes through the climatological controls of the isotopic composition of precipitation."*

*In your study, you present results covering the seasonal and interannual scale. The generalisation to larger time scale is not shown, but hypothesised.*

*Also, in your manuscript, I was under the belief that you considered ice cores as precipitation (and diffusion which you are correcting). A discussion on the differences between precipitation and ice core would be interesting, but I don't believe it is central to your manuscript."*

- We agree that taking the step beyond the seasonal and interannual timescale is not demonstrated in this study, but in section 4.1 we want to show some potential applications of the type of analysis presented in this manuscript for others to examine. We thus think it is appropriate to keep the thought experiments presented in 4.1 in the manuscript.
- Exploring the differences between what is recorded in precipitation and in the ice core would be interesting to examine, but as the reviewer stated in the comment, is not central to this manuscript and beyond the scope of the study. An interesting follow-up project would certainly

be to see how the signal of this precipitation record is preserved in the ice in a follow up study (or studies) after the snow has had time to diffuse.

*"Page 14 – Line 2: "Using the slope of this line adds a new method of inquiry as it holds more information, or at least different information, than simply taking the average δD, δ18O, or d-excess over a given time window."*

*d-excess and the slope δD vs δ18O is the same climatic information."*

- Taking this logic further, one could also say that d-excess also has the same climate information as just δD and $\delta^{18}$O since both the slope and the d-excess are calculated from those values. However, there are certainly additional pieces of information you can learn when looking at d-excess and slope, especially as the slope provides a means to integrate a variety of information about d-excess, δD, and $\delta^{18}$O together. For example, the slope integrates the annual cycles of d-excess and $\delta^{18}$O into a value that can be compared across time (such as what we did in this study) and space.

*"Page 14 – Line 11: "In order to use δD-δ18O slope measurements in ice cores, it is critical to account for the effects of diffusion, which we show to have significant impacts on the slope. If done properly, the method we developed can be applied to deeper cores and/or at other locations."*

*I also believe this is very important. I suggest you include tests that evaluate what impact the correction of effects of diffusion has on the slopes."*

- This has been studied before, by Johnsen et al. (2000), for example. Generally, the slope increases with diffusion, and therefore the d-excess- $\delta^{18}$O phase changes (as pointed out by the other reviewer). Here, we used a published method to correct for diffusion, but the goal is not to study the effect of diffusion itself.

*"Page 14 – Line 33: "Second, while the precise mass balance computations are beyond the scope of this study, the fact that this moisture source significantly contributes to summer precipitation shows that moisture recycling is potentially an important component to consider for the mass balance of the Greenland Ice Sheet."*

*This is indeed out of the scope of the study and brings a lot of questions:*
*- How much moles of water does your result suggest this represent?*
*- What is the relative proportion that this represent compared to the summer accumulation?*
*- How do you distinguish surface sublimation from sublimation of the snow flakes by katabatic winds in coastal areas (Grazioli et al., 2017)? Indeed, the sublimation of the later will not contribute to the SMB.*

*I suggest to remove this sentence or to go in more details."*

- The first two questions are discussed to some degree in Kopec et al. (2019). We can cite that manuscript in this sentence to point readers to those calculations. Regarding the third question,

this is much more complicated to parse out. So long as the vapor sublimation from the snow surface or from falling/blowing snow flakes has a relatively high d-excess, as it is expected to have, it would be difficult with this analysis to tease those apart. However, so long as the moisture is returning to the ice sheet via precipitation at Summit, both scenarios yield net zero SMB changes, and both of which are lesser understood components of mass balance calculations.

*"Page 15 – Line 30: "Over the measurement period of the Owen ice core, the reduction of sea ice has caused an increase of Arctic sourced moisture at Arctic coastal sites (Kopec et al., 2016). If this sea ice effect has also reached Summit, Greenland, we would expect to see the δD-δ18O slope decrease over time. However, after correcting for diffusion, there is no significant temporal trend in the δD-δ18O slope (Fig. 3)."*

*At this stage, I don't think you can reach such conclusion without evaluating the impact of the diffusion correction on your data and by using a single ice core while multiple previous studies have proposed alternative explanations which fit several ice core, and even sometimes [18]O-excess."*

- As stated above, the relationships presented in the multiple regression are reasonably robust to varying diffusion corrections, and thus we believe it is reasonable to present this thought experiment as an example of a potential application of this work.
- Including [17]O-excess into this examination would have been great but that information is not in our data set.

*"References cited by Reviewer 2:*
*Barlow, L., White, J., Barry, R., Rogers, J., and Grootes, P.: The North Atlantic oscillation signature in deuterium and deuterium excess signals in the Greenland Ice Sheet Project 2 ice core, 1840–1970, Geophysical Research Letters, 20, 2901-2904, 1993.*

*Berkelhammer, M., Noone, D. C., Steen-Larsen, H. C., Bailey, A., Cox, C. J., O'Neill, M. S., Schneider, D., Steffen, K., and White, J. W.: Surface-atmosphere decoupling limits accumulation at Summit, Greenland, Science Advances, 2, e1501704, 2016.*

*Casado, M., Landais, A., Masson-Delmotte, V., Genthon, C., Kerstel, E., Kassi, S., Arnaud, L., Picard, G., Prie, F., Cattani, O., Steen-Larsen, H. C., Vignon, E., and Cermak, P.: Continuous measurements of isotopic composition of water vapour on the East Antarctic Plateau, Atmos. Chem. Phys., 16, 8521-8538, 10.5194/acp-16-8521-2016, 2016.*

*Dansgaard, W.: Stable isotopes in precipitation, Tellus, 16, 436-468, 10.1111/j.2153-3490.1964.tb00181.x, 1964.*

*Dittmann, A., Schlosser, E., Masson-Delmotte, V., Powers, J. G., Manning, K. W., Werner, M., and Fujita, K.: Precipitation regime and stable isotopes at Dome Fuji, East Antarctica, Atmospheric Chemistry and Physics, 16, 6883-6900, 2016.*

Ekaykin, A. A., Lipenkov, V. Y., Barkov, N. I., Petit, J. R., and Masson-Delmotte, V.: Spatial and temporal variability in isotope composition of recent snow in the vicinity of Vostok station, Antarctica: implications for ice-core record interpretation, Annals of Glaciology, 35, 181-186, 10.3189/172756402781816726, 2002.

Grazioli, J., Madeleine, J.-B., Gallée, H., Forbes, R. M., Genthon, C., Krinner, G., and Berne, A.: Katabatic winds diminish precipitation contribution to the Antarctic ice mass balance, Proceedings of the National Academy of Sciences, 201707633, 2017.

Hendricks, M., DePaolo, D., and Cohen, R.: Space and time variation of δ18O andδD in precipitation: Can paleotemperature be estimated from ice cores?, Global Biogeochemical Cycles, 14, 851-861, 2000.

Hoffmann, G., Jouzel, J., and Johnsen, S.: Deuterium excess record from central Greenland over the last millennium: Hints of a North Atlantic signal during the Little Ice Age, Journal of Geophysical Research: Atmospheres, 106, 14265-14274, 2001.

Jouzel, J., Hoffmann, G., Koster, R., and Masson, V.: Water isotopes in precipitation:: data/model comparison for present-day and past climates, Quat. Sci. Rev., 19, 363-379, 2000.

Landais, A., Steen-Larsen, H. C., Guillevic, M., Masson-Delmotte, V., Vinther, B., and Winkler, R.: Triple isotopic composition of oxygen in surface snow and water vapor at NEEM (Greenland), Geochimica et Cosmochimica Acta, 77, 304-316, 2012.

Landais, A., Casado, M., Prié, F., Magand, O., Arnaud, L., Ekaykin, A., Petit, J.-R., Picard, G., Fily, M., and Minster, B.: Surface studies of water isotopes in Antarctica for quantitative interpretation of deep ice core data, Comptes Rendus Geoscience, 2017.

Masson-Delmotte, V., Hou, S., Ekaykin, A., Jouzel, J., Aristarain, A., Bernardo, R. T., Bromwich, D., Cattani, O., Delmotte, M., Falourd, S., Frezzotti, M., Gallée, H., Genoni, L., Isaksson, E., Landais, A., Helsen, M. M., Hoffmann, G., Lopez, J., Morgan, V., Motoyama, H., Noone, D., Oerter, H., Petit, J. R., Royer, A., Uemura, R., Schmidt, G. A., Schlosser, E., Simões, J. C., Steig, E. J., Stenni, B., Stievenard, M., van den Broeke, M. R., van de Wal, R. S. W., van de Berg, W. J., Vimeux, F., and White, J. W. C.: A Review of Antarctic Surface Snow Isotopic Composition: Observations, Atmospheric Circulation, and Isotopic Modeling*, J. Clim., 21, 3359-3387, 10.1175/2007JCLI2139.1, 2008.

Masson-Delmotte, V., Steen-Larsen, H., Ortega, P., Swingedouw, D., Popp, T., Vinther, B., Oerter, H., Sveinbjörnsdottir, A., Gudlaugsdottir, H., and Box, J.: Recent changes in north-west Greenland climate documented by NEEM shallow ice core data and simulations, and implications for past-temperature reconstructions, 2015.

NEEM: Eemian interglacial reconstructed from a Greenland folded ice core, Nature, 493, 489-494, http://www.nature.com/nature/journal/v493/n7433/abs/nature11789.html#supplementary-information, 2013.

NorthGRIP: High-resolution record of Northern Hemisphere climate extending into the last interglacial period, Nature, 431, 147-151, http://www.nature.com/nature/journal/v431/n7005/suppinfo/nature02805_S1.html, 2004.

Oyabu, I., Matoba, S., Yamasaki, T., Kadota, M., and Iizuka, Y.: Seasonal variations in the major chemical species of snow at the South East Dome in Greenland, Polar Science, 10, 36-42, https://doi.org/10.1016/j.polar.2016.01.003, 2016.

Persson, A., Langen, P. L., Ditlevsen, P., and Vinther, B. M.: The influence of precipitation weighting on interannual variability of stable water isotopes in Greenland, Journal of Geophysical Research: Atmospheres, 116, 2011.

Pfahl, S., and Sodemann, H.: What controls deuterium excess in global precipitation?, Climate of the Past, 10, 771-781, 2014.

Risi, C., Landais, A., Winkler, R., and Vimeux, F.: Can we determine what controls the spatio-temporal distribution of d-excess and 17O-excess in precipitation using the LMDZ general circulation model?, Clim. Past, 9, 2173-2193, 10.5194/cp-9-2173-2013, 2013.

Ritter, F., Steen-Larsen, H. C., Werner, M., Masson-Delmotte, V., Orsi, A., Behrens, M., Birnbaum, G., Freitag, J., Risi, C., and Kipfstuhl, S.: Isotopic exchange on the diurnal scale between near-surface snow and lower atmospheric water vapor at Kohnen station, East Antarctica, The Cryosphere Discuss., 2016, 1-35, 10.5194/tc-2016-4, 2016.

Schotterer, U., Oldfield, F., and Fröhlich, K.: GNIP. Global Network for Isotopes in Precipitation, 1996.

Sodemann, H., Masson-Delmotte, V., Schwierz, C., Vinther, B. M., and Wernli, H.: Interannual variability of Greenland winter precipitation sources: 2. Effects of North Atlantic Oscillation variability on stable isotopes in precipitation, Journal of Geophysical Research: Atmospheres, 113, 2008.

Sokratov, S. A., and Golubev, V. N.: Snow isotopic content change by sublimation, Journal of Glaciology, 55, 823-828, 10.3189/002214309790152456, 2009.

Steen-Larsen, H. C., Masson-Delmotte, V., Sjolte, J., Johnsen, S. J., Vinther, B. M., Bréon, F. M., Clausen, H. B., Dahl-Jensen, D., Falourd, S., Fettweis, X., Gallée, H., Jouzel, J., Kageyama, M., Lerche, H., Minster, B., Picard, G., Punge, H. J., Risi, C., Salas, D., Schwander, J., Steffen, K., Sveinbjörnsdóttir, A. E., Svensson, A., and White, J.: Understanding the climatic signal in the water stable isotope records from the NEEM shallow firn/ice cores in northwest Greenland, Journal of Geophysical Research: Atmospheres, 116, n/a-n/a, 10.1029/2010JD014311, 2011.

Steen-Larsen, H. C., Johnsen, S. J., Masson-Delmotte, V., Stenni, B., Risi, C., Sodemann, H., Balslev-Clausen, D., Blunier, T., Dahl-Jensen, D., Ellehøj, M. D., Falourd, S., Grindsted, A., Gkinis, V., Jouzel, J., Popp, T., Sheldon, S., Simonsen, S. B., Sjolte, J., Steffensen, J. P., Sperlich, P., Sveinbjörnsdóttir, A. E., Vinther, B. M., and White, J. W. C.: Continuous monitoring of summer surface water vapor isotopic

*composition above the Greenland Ice Sheet, Atmos. Chem. Phys., 13, 4815-4828, 10.5194/acp-13-4815-2013, 2013.*

*Steen-Larsen, H. C., Masson-Delmotte, V., Hirabayashi, M., Winkler, R., Satow, K., Prié, F., Bayou, N., Brun, E., Cuffey, K. M., Dahl-Jensen, D., Dumont, M., Guillevic, M., Kipfstuhl, S., Landais, A., Popp, T., Risi, C., Steffen, K., Stenni, B., and Sveinbjörnsdottír, A. E.: What controls the isotopic composition of Greenland surface snow?, Clim. Past, 10, 377-392, 10.5194/cp-10-377-2014, 2014.*
*Steen-Larsen, H. C., Masson-Delmotte, V., Sjolte, J., Johnsen, S. J., Vinther, B. M., Bréon, F. M., Clausen, H., Dahl-Jensen, D., Falourd, S., and Fettweis, X.: Understanding the climatic signal in the water stable isotope records from the NEEM shallow firn/ice cores in northwest Greenland, Journal of Geophysical Research: Atmospheres, 116, 2011.*

*Stenni, B., Scarchilli, C., Masson-Delmotte, V., Schlosser, E., Ciardini, V., Dreossi, G., Grigioni, P., Bonazza, M., Cagnati, A., Karlicek, D., Risi, C., Udisti, R., and Valt, M.: Three-year monitoring of stable isotopes of precipitation at Concordia Station, East Antarctica, The Cryosphere, 10, 2415-2428, 10.5194/tc-10-2415-2016, 2016.*

*Touzeau, A., Landais, A., Stenni, B., Uemura, R., Fukui, K., Fujita, S., Guilbaud, S., Ekaykin, A., Casado, M., Barkan, E., Luz, B., Magand, O., Teste, G., Le Meur, E., Baroni, M., Savarino, J., Bourgeois, I., and Risi, C.: Acquisition of isotopic composition for surface snow in East Antarctica and the links to climatic parameters, The Cryosphere, 10, 837-852, 10.5194/tc-10-837-2016, 2016.*

*Werner, M., Heimann, M., and Hoffmann, G.: Isotopic composition and origin of polar precipitation in present and glacial climate simulations, Tellus B: Chemical and Physical Meteorology, 53, 53-71, 10.3402/tellusb.v53i1.16539, 2001.*

*Werner, M., Jouzel, J., Masson-Delmotte, V., and Lohmann, G.: Reconciling glacial Antarctic water stable isotopes with ice sheet topography and the isotopic paleothermometer, Nature communications, 9, 3537, 2018."*